# Spermine modulation of Alzheimer's Tau and Parkinson's α-synuclein: implications for biomolecular condensation and neurodegeneration

Xun Sun[1], Debasis Saha[2], Xue Wang[1], Cecilia Mörman[1,3], Rebecca Sternke-Hoffmann [1], Juan Atilio Gerez [4], Fátima Herranz-Trillo[5], Roland Riek [4], Wenwei Zheng [2] & Jinghui Luo [1]✉

Spermine, a pivotal player in biomolecular condensation and diverse cellular processes, has emerged as a focus of investigation in aging, neurodegeneration, and other diseases. Despite its significance, the mechanistic details of spermine remain incompletely understood. Here, we describe the distinct modulation by spermine on Alzheimer's Tau and Parkinson's α-synuclein, elucidating their condensation behaviors in vitro and in vivo. Using biophysical techniques including time-resolved SAXS and NMR, we trace electrostatically driven transitions from atomic-scale conformational changes to mesoscopic structures. Notably, spermine extends lifespan, ameliorates movement deficits, and restores mitochondrial function in *C. elegans* models expressing Tau and α-synuclein. Acting as a molecular glue, spermine orchestrates in vivo condensation of α-synuclein, influences condensate mobility, and promotes degradation via autophagy, specifically through autophagosome expansion. This study unveils the interplay between spermine, protein condensation, and functional outcomes, advancing our understanding of neurodegenerative diseases and paving the way for therapeutic development.

As the global population ages, the incidence of age-related and neurodegenerative diseases rises accordingly. Progressive neuronal dysfunction and accumulation of amyloid fibrils are the hallmark of various neurological disorders, including Alzheimer's disease (AD)[1], and Parkinson's disease (PD)[2]. Along with amyloid protein fibril deposition, other contributing factors to these neurodegenerative diseases including epigenetic modifications[3], genomic stability[4], oxidative stress[5], neuroinflammation, mitochondrial dysfunction[6], and a defective autophagy system[7], underscore the endogenous complexity of the diseases. A comprehensive understanding of the endogenous factors that impede amyloid protein deposition and interact with diverse biomolecules is crucial for developing anti-aging therapies and effective treatments for neurodegenerative disorders.

Endogenous polyamines, particularly spermine, have garnered significant attention for their potential roles in modulating neurodegenerative process[8]. Polyamines, including spermine, spermidine, and putrescine, are known to interact with nucleic acids and proteins, playing integral roles in cellular homeostasis and regulation pathways[9]. Of particular interest, spermine has been reported to confer neuroprotection against age-related memory decline and mitigate α-

[1]Center for Life Sciences, Paul Scherrer Institute, Villigen, PSI, Switzerland. [2]College of Integrative Sciences and Arts, Arizona State University, Mesa, AZ, USA. [3]Department of Medicine Huddinge, Division of Biosciences and Nutrition, Karolinska Institutet, Huddinge, Sweden. [4]Institute of Molecular Physical Science, Department of Chemistry and Applied Biosciences, ETH Zurich, Zurich, Switzerland. [5]CoSAXS Beamline, MAX IV Laboratory, Lund, Sweden. ✉e-mail: jinghui.luo@psi.ch

synuclein (αS)-induced neurotoxicity in animal models[10,11]. Additionally, both spermidine and spermine have been found to induce autophagy, a critical process that maintains proteostasis by degrading damaged organelles and toxic protein aggregates—an essential defense against neurodegeneration[12]. Numerous studies have demonstrated that autophagy can eliminate Tau and αS aggregates in microglia and neuronal cells[13], presumably with macroautophagy, a selective form of autophagy, relying on the formation of autophagosomes to sequester and degrade protein aggregates[14].

Despite these advances, the mechanisms underlying amyloid aggregate degradation through autophagy in the context of neurodegeneration remain incompletely understood. Biomolecular condensates, formed through liquid-liquid phase separation (LLPS), are emerging as central players in cellular organization[14–16] and may provide insights into this process. Protein condensation likely plays an important role in multiple steps of autophagy, including the assembly of autophagosome formation sites, acting as an p62 condensate-mediated autophagy recognition, and classifying protein cargo for degradation[14]. These dynamic protein condensates, which exist in liquid- or gel-like states, are distinct from insoluble aggregates and are formed and dissolved through complex networks of homotypic and heterotypic interactions[17]. Heterotypic buffering, a critical mechanism, mitigates the deleterious effects of homotypic interactions[17] and may influence pathological LLPS behavior in neurodegenerative diseases.

We hypothesize that liquid-like condensates, as opposed to solid fibrils, are more adaptable to dynamic cellular processes, such as autophagosome formation and expansion. These fluid condensates can concentrate biomolecules at specific cellular locales, facilitating key cellular functions without the hindrance of physical barriers. Small molecules like molecular glues may further regulate the interactions within protein condensates, allowing precise control over cellular dynamics. Spermine, with its polycationic properties, is a promising candidate for modulating the interactions between the amyloidogenic proteins Tau and αS, both of which have been shown to phase separate[18–20], but with distinctive charge patterning within their sequences. While the term molecular glue is often used to describe small molecules that induce protein–protein interactions by directly bridging two components, here we use the term more broadly to describe the ability of spermine to modulate heterotypic interactions and promote co-condensation via charge neutralization. This usage is operational and refers to the enhancement of intermolecular association rather than a defined ternary complex formation. Given spermine's ability to facilitate DNA recombination and condense duplex DNA through electrostatic interactions[21,22], it may similarly influence the condensation of these amyloidogenic proteins and their effects on cellular degradation pathways.

In this study, we explore the potential of spermine as a molecular glue to regulate the conformation, condensation and ultimately degradation pathway of Tau and αS, spanning atomic-level interactions, mesoscopic LLPS behavior in vitro to functional correlates in vivo. Using time-resolved small-angle X-ray scattering (SAXS), nuclear magnetic resonance (NMR), and coarse-grained (CG) molecular dynamics simulations, we illustrate how spermine modulates the conformations of Tau and αS via electrostatic interactions. We then characterize the LLPS behaviors of Tau and αS using light microscopy, followed by in vivo studies in a *C. elegans* model expressing these proteins. Our findings suggest that spermine promotes αS condensation and increases the mobility of cellular droplets, potentially linked to the autophagy pathway and autophagosome expansion.

## Results
### Spermine-mediated LLPS of full-length Tau and αS in vitro
The full-length human Tau441 (2N4R) and K18 (Fig. 1A) are inherently disordered and display a unique multivalent pattern in charge distribution under physiological pH 7.4 conditions. The catGranule

method[23] predicts an overall droplet-promoting propensity to undergo LLPS of Tau $p_{LLPS} = 1.689$ and K18 $p_{LLPS} = 2.082$ (Supplementary Fig. 1A, C). Higher $p_{LLPS}$ values indicate a stronger propensity for phase separation, suggesting that K18 exhibits a greater tendency to undergo LLPS compared to Tau. We first aimed to gain insight into the impact of spermine (Supplementary Fig. 2) on the LLPS of Tau and K18 at physiological pH under various concentrations. In the presence of 10% or 7.5% of PEG, 10 μM Tau showed the droplet formation with 10 or 50 μM spermine (Fig. 1B). In contrast, spermine did not alter the LLPS behavior of K18 in the presence of 10% (Supplementary Fig. 3A, B) or 7.5% of PEG (Supplementary Fig. 3C, D). The droplet formation was further confirmed by the fluorescence imaging with Alexa-647 C2 Maleimide-labelled Tau (5% labelled) protein (Fig. 1B). As shown in the phase diagrams of Tau LLPS under 10% PEG (Fig. 1C and Supplementary Fig. 4A) and 7.5% PEG (Supplementary Fig. 4B, C), enhanced molecular crowding reduced the critical concentration of Tau necessary for spermine-induced phase separation.

To study the maturation of K18 and Tau droplets, the time-lapse images were acquired by using a protein crystallization robotic imager. We did not observe any difference of the droplet starting point and size of 10 μM K18 with or without 100 μM spermine (Supplementary Fig. 5A, B). However, spermine accelerated 20 μM Tau droplet formation compared to Tau alone (Supplementary Fig. 6A, B), although the size of all droplets increased with time. To study the dynamic mobility of K18 and Tau LLPS, droplet fusion event, the coalescence of two small droplets to form a larger droplet, was observed in both K18 and Tau LLPS with or without spermine (Fig. 1E & Supplementary Fig. 5C). To characterize the dynamics of the Tau proteins inside liquid droplet, we performed fluorescence recovery after photobleaching (FRAP) experiment using droplet formed with Alexa-647 labeled Tau. A complete Tau fluorescence recovery (~95%) was observed in the presence of spermine compared to ~26% fluorescence recovery of Tau alone, which indicated that spermine increased the mobility of Tau droplet (Fig. 1D & Fig. S7A). To further test the droplet property, we explored the impact of 1,6-hexanediol, a known LLPS inhibitor for intrinsically disordered proteins (IDPs), likely by disrupting hydrophobic interactions[24]. In the presence of 10% 1,6-hexanediol, Tau droplets formed without spermine were completely dissolved, while those formed with spermine exhibited partial resistance to dissolution, and the number of droplets was reduced (Fig. 1F). This suggests that although spermine promotes droplet formation likely through electrostatic interactions, hydrophobic interactions remain essential for the overall phase behavior. This is evidenced by the continued, though attenuated, sensitivity to 1,6-hexanediol with the presence of spermine, suggesting a likely shift from hydrophobic toward electrostatic interactions.

To examine the effect of spermine on the LLPS of αS, we then investigated whether αS's distinct sequence charge distribution influences spermine's effects on phase separation. In contrast to Tau, the sequence of full-length human αS shown in Fig. 1G, reveal αS as an acidic disordered protein with a net charge of negatively 9.7 at physiological pH 7.4 and a dipolar charge distribution: an amphipathic N-terminal domain (aa 1-60) with a moderate excess of positively 2.3 charges and a hydrophobic central NAC region (aa 61-95) and acidic C-terminal domain (aa 96-140) having negatively 12.8 charges (Fig. 1G). The catGranule method predicts an overall droplet-promoting propensity to undergo LLPS of $p_{LLPS} = 1.130$ (Supplementary Fig. 1B-C), suggesting a substantially lower tendency to undergo phase separation compared to Tau. To investigate the impact of spermine on the LLPS behavior of αS at pH 7.4, LLPS assays were carried out. In the presence of 10% or 7.5% of PEG, 20 μM αS droplets formation was observed in the presence of 10 or 50 μM spermine (Fig. 1H). The droplet formation was further confirmed by the fluorescence imaging with Alexa-647 C2 Maleimide-labelled αS (5% labelled) protein (Fig. 1H). Moreover, the phase diagrams of αS LLPS in the presence of 10% PEG

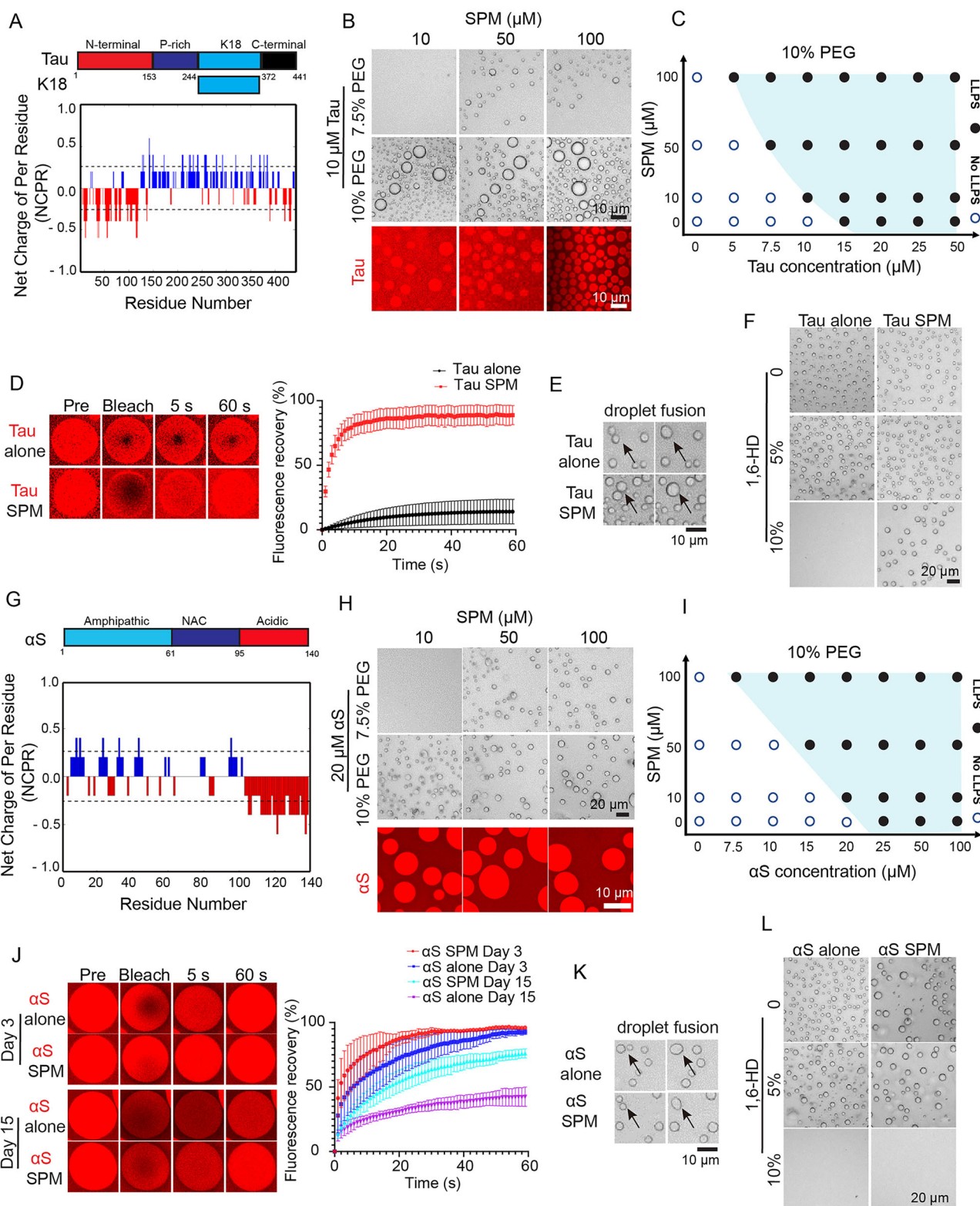

(Fig. 1I & Supplementary Fig. 8A) and 7.5% PEG (Supplementary Fig. 8B, C) indicate that increasing molecular crowding lowers the critical concentration required for spermine-mediated α-synuclein phase separation.

To study the maturation of αS droplet, we found that spermine accelerated 25 µM αS droplet formation (Supplementary Fig. 9A, B). To assess the dynamic mobility of αS during LLPS, droplet fusion events

were observed in 25 µM αS samples in the absence and presence of 100 µM spermine (Fig. 1K). Furthermore, to characterize the internal dynamics of the αS proteins within the liquid droplet, we performed FRAP experiment with Alexa-647 labeled αS droplets collected on days 3 and 15 (Fig. 1J & Supplementary Fig. 7B). Spermine strongly increased αS fluorescence recovery on days 3 and 15. Treatment of 1,6-Hexanediol, used to assess hydrophobic interactions, completely

**Fig. 1 | Effects of spermine on human full-length Tau and αS LLPS in vitro. A** Up panel: full-length human Tau (2N4R) protein sequence. The highly disordered N-terminal domain (red), proline-rich regions (dark blue), K18 domain (light blue) and C-terminal domain (black). Below panel: Single amino acid charge distribution along the Tau sequence (at pH 7.4) were calculated at the CIDER server[73]. It consists of the negatively charged (red) N-terminal domain, the positively charged (blue) middle domain, and the negatively charged (red) C-terminal domain. **B** Up panel: representative images of 10 μM Tau in presence of increasing concentration of spermine and 7.5% or 10% of PEG. Below panel: fluorescence images of Alexa-647 labelled Tau (10 μM) phase-separated droplets formation in the presence of increasing concentration of spermine and 10% of PEG. **C** phase diagram corresponding to images (Supplementary Fig. 4A) illustrating the phase separation of Tau at different concentrations of protein and spermine in the presence of 10% PEG. **D** Representative droplet images FRAP measurement of 20 μM Tau-Alexa-647 in the absence and presence of 100 μM spermine (n = 3). **E** liquid-like droplet fusion of 20 μM of Tau in the absence and presence of 100 μM spermine and 10% PEG. **F** Effects of different percentage of 1,6-HD treatment on 20 μM of Tau in the absence and presence of 100 μM spermine in the presence of 10% PEG. **G** Up panel: full-length human αS protein sequence. The amphipathic N-terminal domain (light blue), highly hydrophobic central region (dark blue) and acidic C-terminal domain (red). Below panel: Single amino acid charge distribution along αS sequence (at pH 7.4). The sequence contains the negatively charged (red) C-terminal domains, the positive charged (blue) N-terminal and middle domains. **H** Up panel: representative images of 20 μM αS in presence of increasing concentration of spermine and 7.5% or 10% of PEG. Below panel: fluorescence images of Alexa-647 labelled αS (20 μM) phase-separated droplets formation in the presence of increasing concentration of spermine and 10% of PEG. **I** phase diagram corresponding to images (Supplementary Fig. 8A) illustrating the phase separation of αS at different concentration of protein and spermine in the presence of 10% PEG. **J** Representative droplet images FRAP measurement of 25 μM αS-Alexa-647 in the absence and presence of 100 μM spermine at the indicated time points (n = 3). **K** liquid-like droplet fusion of 25 μM of αS-Alexa-647 in the absence and presence of 100 μM spermine in the presence of 10% PEG. **L** Effects of different percentage of 1,6-HD treatment on 25 μM of αS in the absence and presence of 100 μM spermine and 10% PEG. Data are presented as mean values +/− SEM. Representative microscopy images are from at least three independent biological replicates, unless indicated otherwise.

dissolved αS droplets, in both the presence and absence of spermine (Fig. 1L).

To further investigate the impact of spermine on Tau and αS fibrillization within condensates, we performed ThT fluorescence kinetics assay. As shown in Supplementary Fig. 10, spermine reduced both Tau and αS fibrillization in a dose-dependent manner. For Tau, spermine increased the lag time and half-time of aggregation while decreasing the final ThT intensity, indicating delayed and reduced fibril formation (Supplementary Fig. 10A–D). Similarly, spermine profoundly inhibited αS aggregation, as shown by the markedly reduced ThT plateau values at higher spermine concentrations (Supplementary Fig. 10E–H). These results demonstrate that spermine acts as an inhibitor of amyloid fibril formation for both Tau and αS under molecular crowding conditions.

Overall, Tau and K18 exhibit distinct behaviors with spermine, as Tau undergoes LLPS in the presence of spermine, forming droplets at physiological pH, whereas K18 shows no such response to spermine. In contrast, αS, like Tau but with additional charged residues, forms droplets in the presence of spermine. However, its acidic and amphipathic sequence results in different charge distributions and LLPS characteristics. Additionally, spermine increases the dynamics and mobility of droplets in both Tau and αS, as observed by FRAP, highlighting the importance of further investigating the interplay between spermine and protein sequence in LLPS to inform the design of small molecular glues.

## Global and local conformations of Tau and αS in the presence of spermine

To further investigate the role of spermine on LLPS of Tau and αS, global conformational dynamics of Tau and αS in the presence of spermine were probed via time-resolved small-angle X-ray scattering (TR-SAXS). The schematics of the microfluidic device has been shown in Fig. 2A. Experimental TR-SAXS curves of Tau alone and Tau with spermine are presented in Supplementary Fig. 11A, B, respectively. The Ensemble Optimization Method (EOM)[25] was applied to fit the experimental data in red. The Tau concentration was fixed to 100 μM, and the microfluidic device allowed measurements from 70 ms. These measurements revealed the $Rg$ (radius of gyration) is roughly 55 Å over the measured time for Tau, with a slight decrease observed over the measurement time from 56.5 Å (0.07 s) to 53.3 Å (0.92 s), likely attributable to the flexibility of IDP. Employing the same setup and microfluidic device, we recorded the response of Tau to additional spermine within the same measurement time window, ensuring proper device functionality and attributing structural conformational changes solely to the presence of additional spermine. However, upon addition of spermine, Tau rapidly compacted within ~0.2 seconds, as indicated by a sharp decrease in $Rg$, followed by a secondary expansion into a globular conformation with an $Rg$ of 68.9 Å (Fig. 2B). This suggests that the subsequent conformational rearrangement may be governed by intra- and intermolecular interactions, limiting additional compaction.

Given the distinct charge distributions between αS and Tau sequences, as well as their enhanced droplet formation behaviors induced by spermine, we employed an identical experimental setup and microfluidic device to explore how spermine influences the structural changes of αS at the initial stage (Supplementary Fig. 11C, D). Our results revealed that the $Rg$ of αS alone remained constant throughout the measurement duration (black lines in Fig. 2C), indicating the stability of monomeric αS within the microfluidic system. In contrast, Tau and αS displayed distinct conformational responses to spermine (red lines in Fig. 2B, C). Tau underwent rapid initial compaction followed by an increase in its $Rg$ (Fig. 2B), while the subsequent expansion may be driven by weak intermolecular attractions and intramolecular repulsions. To rule out early-stage aggregation as the cause of the $Rg$ increase, mass photometry confirmed that Tau remained predominantly monomeric (~45 kDa) with no detectable aggregates (Supplementary Fig. 12), and ThT assays also supported the absence of aggregation within the timescale of our TR-SAXS measurements (Supplementary Fig. 10). In comparison, αS exhibited a gradual and steady increase in $Rg$ without an initial collapse, suggesting that intra- and inter-molecular interactions consistently influence the size of αS. These divergent global dynamics may be attributed to differences in their residue-level interactions with spermine, resulting from distinct charge distributions.

To shed more light on the potential residue-specific interactions between monomeric Tau and spermine, NMR spectroscopy experiments were performed in the PBS buffer with a pH 6.3. Spermine was added onto 100 μM $^{15}$N-labeled Tau protein and HMQC spectra before and after 1 mM spermine addition were recorded and compared (Fig. 2D). As shown in Supplementary Fig. 13, spermine-induced phase separation in PBS buffer displays qualitatively similar behavior to that observed in HEPES buffer (Fig. 1B). Tau exhibits a spectrum resembling of a large unstructured protein with typical features such as only partial dispersed chemical shifts, which are contributing to overlap of several amide crosspeaks. Interestingly, in the presence of spermine, several, but not all, well-defined crosspeaks corresponding to a few specific amino acids in the Tau primary sequence displayed both a loss of intensity and chemical shift changes (Fig. 2D & Supplementary Fig. 14A). Both effects, intensity reduction and chemical shift change, indicate an interaction between Tau and spermine. However, despite a general intensity loss of approximately 20 to 60%, particularly more pronounced in the N- and C-terminal regions, no clear binding site(s) are observed. Instead, a diffusive effect is seen across several residues

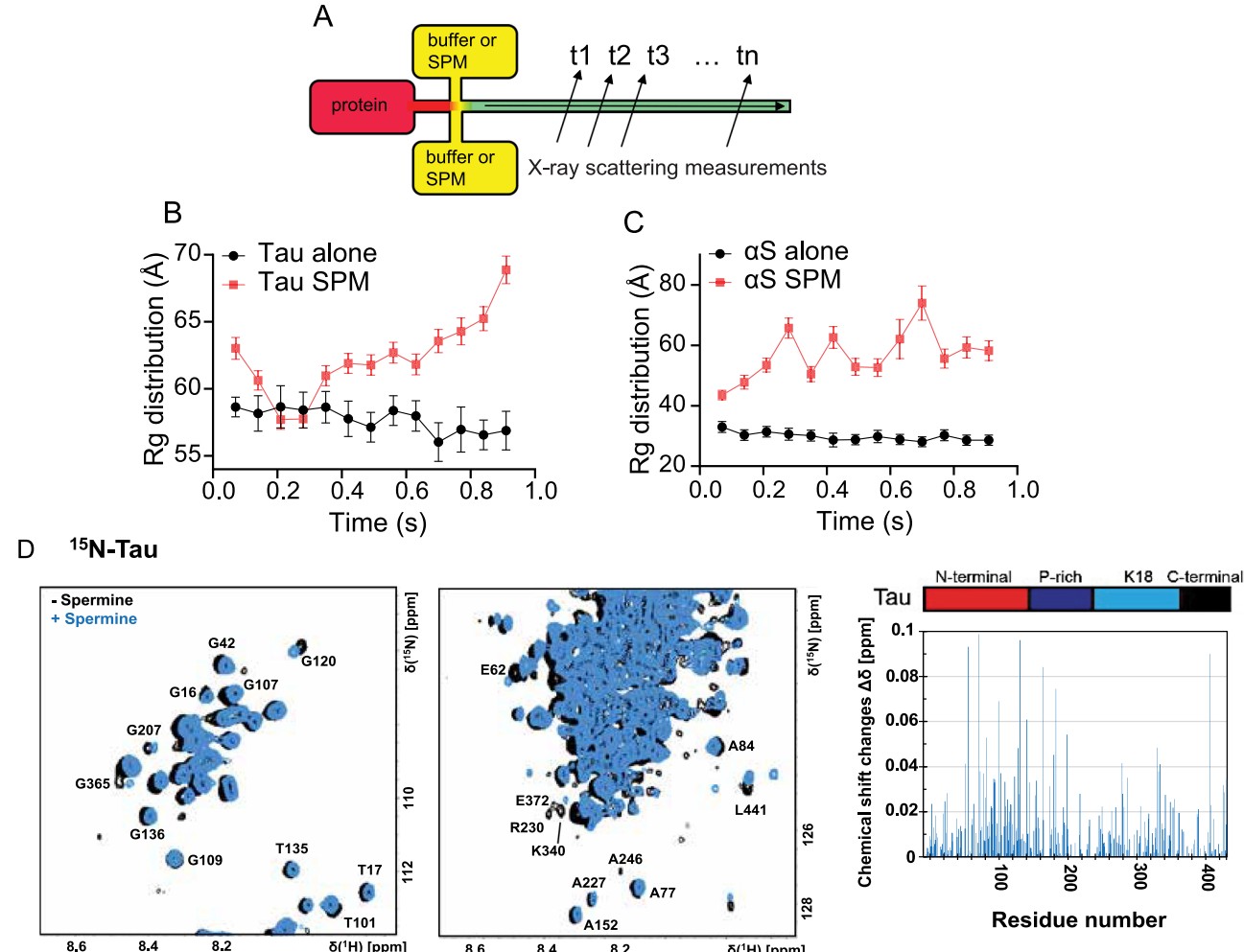

**Fig. 2 | TR-SAXS and 2D 1H-15N-HMQC NMR showing the primary condensation kinetics and interactions between Tau and αS with spermine. A** Schematic illustration of microfluidic device with 3 channels. **B** the $Rg$ distribution of Tau alone and Tau with spermine within 0.92 s ($n = 50$). **C** the $Rg$ distribution of αS alone and αS with spermine within 0.92 s ($n = 50$). **D** Spectra of $^{15}$N-Tau in the absence and presence of spermine in PBS buffer pH 6.3. In the right panel is the chemical shift changes from the spectra presented as a function of the residue number in the protein primary sequence. The spectra were recorded at 283 K. Full spectra are shown in Figure S14. Data are presented as mean values +/− SEM.

within Tau. Interestingly, from the chemical shift changes a more pronounced effect seems to be in the negatively charged N-terminal region of Tau, suggesting the importance electrostatic interactions with spermine.

To gain further information, 100 μM of the shorter K18 Tau fragment ($^{15}$N-labeled), with more well-resolved amide crosspeaks in the HMQC spectrum compared to the full-length Tau protein, was used (Supplementary Fig. 14B & Supplementary Fig. 15). Noteworthy, beyond the differences in primary sequence length between Tau and K18, the charge distribution and total positive net charges are clearly different. Likewise, similar effects were observed for the K18 Tau fragment variant in the presence of spermine as for full-length Tau, suggesting interactions between these residues and spermine. The presence of spermine affects several residues, in particular charged residues. Some of the crosspeaks lose all signal intensity and are completely gone, such as D252 and K340. Residue E342 shows only 20% of the signal intensity left after addition of spermine. One of the largest displacements of the chemical shifts in the K18 Tau protein is D348. Another observation includes less of a decrease of the crosspeak signal intensities for the shorter K18 fragment compared to the full-length Tau protein, which supports the suggestion of substantial electrostatic interactions (Supplementary Fig. 14C, D). Taken together,

this high-resolution structural information brings insights into how spermine interacts with Tau at the monomeric level.

In contrast, the previous work on αS highlights more defined electrostatic interactions, where αS undergoes conformational changes in the presence of spermine[26], particularly in its acidic C-terminal region, which is rich in negatively charged residues. These differences suggest that Tau interacts with spermine in a more distributed, weak-binding fashion, while αS may form more specific electrostatic contacts, particularly involving its acidic domains, contributing to its unique aggregation and folding behavior. These residue-specific interaction differences provide a molecular basis for understanding the distinct SAXS and LLPS behaviors of Tau and αS in the presence of spermine, highlighting the potential role of specific and nonspecific interactions in modulating protein global conformations and their resulting phase behavior.

**Spermine mediated conformational changes of Tau and αS follow divergent interaction patterns**

In order to further understand how local and global conformational properties convey to the macroscopic properties, we utilize an existing residue-based coarse-grained (CG) model[27,28] (see Methods) to simulate both single-chain and LLPS properties. The coarse-grained model

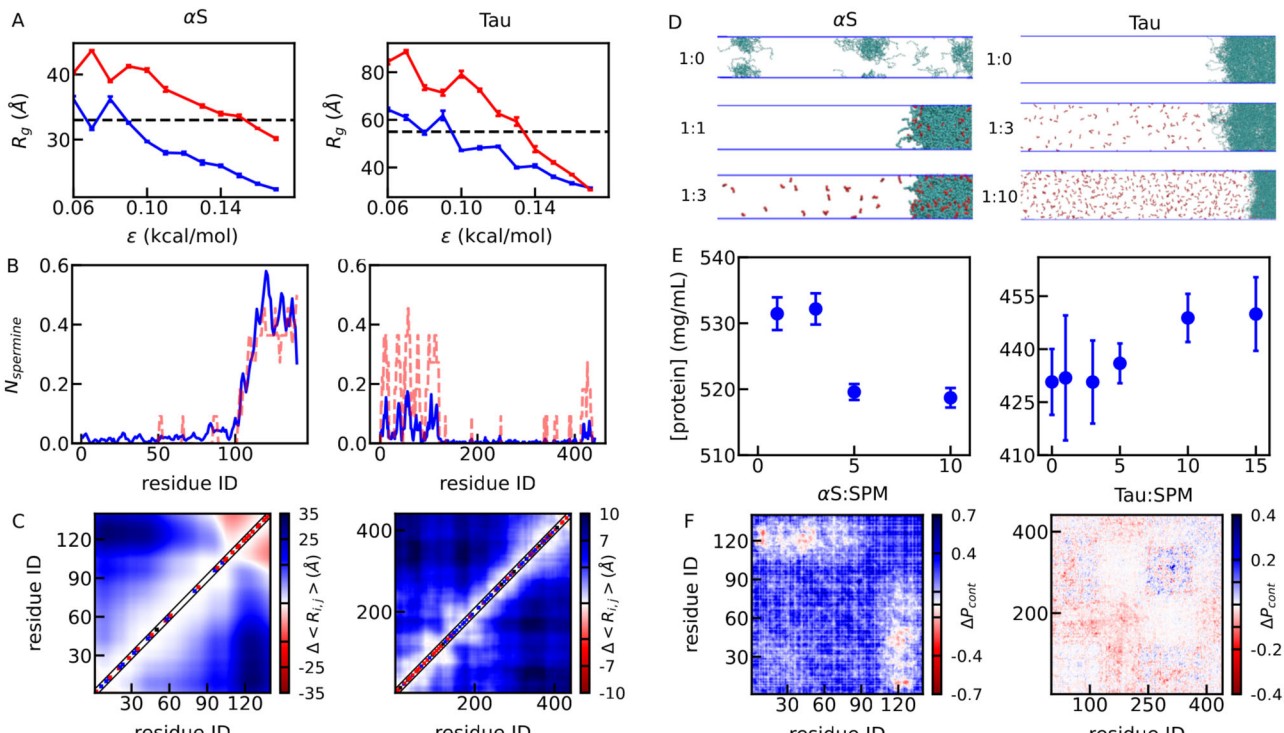

**Fig. 3 | Coarse-grained simulations of αS and Tau investigating role of charged amino acids. A** Radius of gyration of αS and Tau with (red) and without spermine (blue) at different pairwise hydrophobic interaction strengths ε. **B** Number of spermine molecules within contact distance of each amino acid (blue) and averaging net charge per residue calculated using a window size of nine amino acids (red). **C** Difference of pairwise amino acid distances between simulations with and without spermine. **D** Conformations in co-existence slab simulations at different protein-spermine mixing ratios. Note that only half of the symmetric slab simulation box is shown. **E** Concentrations of the high-density phase at different protein-spermine mixing ratios. **F** Difference of intermolecular contact probabilities with and without spermine in coexistence simulations. For (**A**, **E**), data are presented as mean values, with error bars representing the SEM estimated from four technical replicates obtained by block averaging of the 4 μs production trajectory ($n = 4$).

was previously used to capture the salt-dependent phase separation of αS[28]. Here, one spermine molecule is represented as four beads, and every bead has a charge of +1. We first probe the impact of spermine on the $R_g$ of αS and Tau using single-chain simulations. As shown in Fig. 3A, we scanned a variety of hydrophobic interaction strengths, denoted by ε, which controls the strength of short-range pairwise interactions between residues. While ε is a model parameter, it is typically calibrated to reproduce experimental observables such as $R_g$, and trends are confirmed across a range of values to ensure robustness. Using TR-SAXS experiment as guidance, we found an ε value of approximately 0.09 kcal/mol best match the $R_g$ of αS and Tau at the experimental ionic strength without spermine.

For Tau, the current CG model does not reproduce spermine-dependent phase separation as observed in the experiment. The co-existence (slab) simulations showed no spermine dependence in condensate formation (Supplementary Fig. 16). Motivated by K18's aggregation-prone nature, we increased the ε only for interactions between hydrophobic amino acids in the K18 region (residues 244–372). Simulations across various interaction strengths show that K18 compaction increases, while the full-length Tau's $R_g$ remains largely unaffected (Supplementary Fig. 17). Based on these results, all simulations of Tau have been performed with an increasing ε of 0.5 kcal/mol for the hydrophobic residues within K18 region.

Consistent with TR-SAXS measurements (Fig. 2B, C), our CG simulations demonstrate spermine-induced expansion of both αS and Tau, as reflected by the increasing $R_g$ values in the presence of spermine. Interestingly, at higher ε values—intended to mimic increased cellular crowding—this expansion effect persists for αS but is diminished for Tau. This difference likely arises from the higher net charge density of αS, which promotes stronger electrostatic binding with

spermine. In contrast, Tau, which adopts a more collapsed conformation under stronger crowding conditions, becomes less responsive to spermine, likely due to reduced accessibility of interaction sites. Beyond net charge, global charge patterning may also influence conformational responses to spermine. We quantified charge patterning using the κ parameter, which measures how mixed or segregated charged residues are along the sequence. Both αS and Tau exhibit low κ values (-0.17 to 0.18), indicating well-mixed charge distributions. To mimic spermine-induced neutralization, we performed in silico substitution of acidic residues with neutral ones and found that κ values remain largely unchanged across substitution levels (Supplementary Fig. 18). These results suggest that charge patterning is unlikely to be the primary factor driving the distinct global conformational responses of αS and Tau to spermine.

To further understand how the local interactions contribute to the global conformations, we calculated the number of spermine molecules around each amino acid during the simulation (Fig. 3). Although lower ε values better reproduce the experimentally observed $R_g$ of Tau and αS, we show the results in Fig. 3B using simulations at ε = 0.15 kcal/mol, as such a stronger interaction strength mimics the crowding condition used in our LLPS simulations and allows for clearer visualization of spermine-residue contacts under more compact conformations. The analysis finds that spermine mostly interacts with the C-terminal tail of αS whereas spermine does not have a strong preference for a specific region of Tau, consistent with the experimental measurements using NMR chemical shift perturbations. To relate these interactions to charge distribution, we calculated the net charge per residue using a sliding window size of nine amino acids (red dashed lines in Fig. 3B). The correlation between local charge and spermine binding suggests that charged residues drive spermine-protein interactions. To probe

conformational consequences, we compared pairwise amino acid distances with and without spermine (Fig. 3C). In αS, spermine reduces distances between the NTD/NAC and C-terminal tail, promoting compaction. In contrast, spermine expands Tau by neutralizing its acidic regions, reducing intramolecular electrostatic attractions, and increasing $R_g$.

We next conducted co-existence slab simulations (see Methods) to investigate the LLPS behaviors. The simulations always started with a slab conformation, condensing all the proteins in the middle of the box. If the protein fails to undergo phase separation under certain conditions (Fig. 3D, αS: spermine=1:0), slab breaks in the simulation. If the condition favors LLPS (other cases in Fig. 3D), there will be co-existence of two different states with different densities. For αS, the simulations correctly reproduced the spermine-induced phase separation. As shown on the left side of Fig. 3D even using a large interaction strength of $\varepsilon = 0.17$ kcal/mol in simulations, αS still does not undergo phase separation in the absence of spermine. However, when αS is mixed with spermine in a 1:1 ratio, clear phase separation begins to occur. The simulation showed that the density reached a maximum when αS-spermine ratio is 1:3 before decreasing at higher spermine concentrations.

For Tau, we first varied the $\varepsilon$ for residues other than the hydrophobic residues in K18 region, for which the $\varepsilon$ has been fixed at 0.5 kcal/mol to run the LLPS simulations. The same protocol has been used for Tau as was used for αS. We found that beyond $\varepsilon = 0.15$ kcal/mol, Tau showed phase separation in our simulations both in the absence and presence of spermine as can be seen in the right panel of Fig. 3D. When a number of Tau-spermine ratio was simulated, we observed that the slab becomes more stabilized, and the protein density in the high-density phase increases as the number of spermine molecules rises in the simulations, as can be seen on the right side of Fig. 3E. To statistically evaluate the trend, we performed a bootstrap analysis using the experimental density values and their errors. The results show that 95% of fitted slopes were positive, with a one-sided 95% confidence lower bound of 0.07 (Supplementary Fig. 19), supporting a spermine-dependent increase in Tau density. Additionally, the need to adjust the model to accurately reproduce spermine-induced phase separation of Tau suggests the presence of residual structure within the K18 region.

It is important to note that the CG simulation results discussed here correspond to the pH 6.3 conditions of the NMR experiments shown in Fig. 2D. We assigned a charge of +0.5 to histidine residues in the simulations, reflecting their partial protonation at pH 6.3 based on their pKa values. However, the in vitro and in vivo assays in our study were conducted at pH 7.4, where histidine residues are expected to be fully deprotonated and thus carry a net charge of 0. This effect is more likely to impact Tau, which contains 12 histidine residues, but is less relevant for αS, which has only one. To assess whether the observations from our simulations remain valid under these conditions, we performed additional simulations of Tau phase separation, both in the absence and presence of spermine, using a histidine charge of 0 for pH 7.4. The condensate densities obtained from these simulations are compared in Supplementary Fig. 19A with those from our original simulations at pH 6.3. The results show that, although the absolute condensate densities shift to higher values at pH 7.4, the qualitative trends, particularly regarding spermine's ability to promote phase separation, remain consistent across the Tau-spermine ratios examined. These findings suggest that while slight acidification in the NMR experiments may influence the magnitude of electrostatic interactions or condensate density, it does not affect the overall conclusions regarding spermine's role in promoting Tau phase separation.

At last, we checked how the intermolecular contact map changes for αS when the protein-spermine ratio changed from 1:1 to 1:3 and for Tau when Tau-spermine ratio changed from 1:3 to 1:10. The intermolecular contact maps shown in Fig. 3F reveal that αS and Tau follow divergent patterns when varying spermine concentrations. For αS, spermine effectively enhances intermolecular interactions between the NTD and NAC. This is primarily due to spermine's specific interactions with the CTT, where it neutralizes the negative charges within the CTT, thereby reducing its attractive interactions with other parts of the protein. In contrast, for Tau, the impact of increasing spermine concentrations on intermolecular interactions between Tau proteins is much less pronounced than in αS. This aligns with the observation that spermine does not primarily interact with a specific region within Tau.

Although both αS and Tau show spermine-induced expansion, the molecular mechanisms driving their LLPS differ. The behavior of αS can be largely explained by the specific interactions between its negatively charged CTT and spermine. The more positively charged coated CTT reduces its intramolecular interactions with NTD and NAC, causing the chain to extend. This, in turn, enhances the intermolecular interactions between NTD and NAC, facilitating LLPS and possibly aggregation. For Tau, while similar spermine-induced expansion and LLPS are observed, spermine primarily functions to neutralize the negatively charged amino acids throughout the entire sequence rather than targeting specific regions. This neutralization reduces the intramolecular electrostatic attractions and leads to molecular expansion. The resulting expansion, combined with diminished electrostatic repulsion between molecules due to charge neutralization, facilitates Tau LLPS by enabling more frequent and permissive intermolecular encounters.

## Spermine-induced phase separation facilitates αS degradation via the autophagy pathway

Building on the previous efforts regarding spermine-mediated phase separation of αS, we aim to further investigate how this process correlates with αS condensation in the presence of spermine in vivo. We utilized a *C. elegans* model expressing αS-YFP[29]. Subsequently, we examined the age-dependent expression patterns of αS under spermine treatment. Aging is a primary factor affecting PD progression, with conditions worsening as individuals age. We conducted time-kinetics experiments to explore age-dependent aspects, extending to the late adult stage of the worms. Using the transgenic worm strain NL5901 treated with 500 μM spermine or control, we monitored αS condensate formation in both control and spermine-treated worms using confocal microscopy at various adulthood stages. Our observations revealed that aging led to increased αS deposition in control worms (Fig. 4A). However, with the spermine treatment, αS accumulation is significantly reduced in comparison to the control group at the same stage of adulthood (Fig. 4A, B & Supplementary Fig. 20).

It remains unclear whether the spermine-induced reduction in αS accumulation observed in vivo is a result of spermine-induced LLPS observed in vitro, as previously shown or not. To further investigate this, we examined spermine's effect on the physical properties of αS condensates over time, specifically the mobility of the αS condensate, using FRAP. We observed that under spermine treatment, the αS condensates retained liquid-like properties up to Day 15 of adulthood (Fig. 4C–E & Supplementary Fig. 21), with rapid fluorescence recovery. In contrast, low fluorescence recovery was observed on Day 7 and Day 15 in the control group without the presence of spermine, indicating that these αS condensates exhibited solid-like properties. Taken together with the previously observed spermine-induced LLPS, the findings from the *C. elegans* model demonstrate that the spermine treatment significantly increased the mobility of αS condensates, promoting LLPS rather than forming solid aggregates. This observation is also consistent with the in vitro FRAP experiments, which showed enhanced mobility of αS condensates in the presence of spermine. This increased mobility confirms spermine's role in promoting the liquid-like properties of αS both in vitro and in vivo, which might facilitate the degradation of αS condensates and require further investigation.

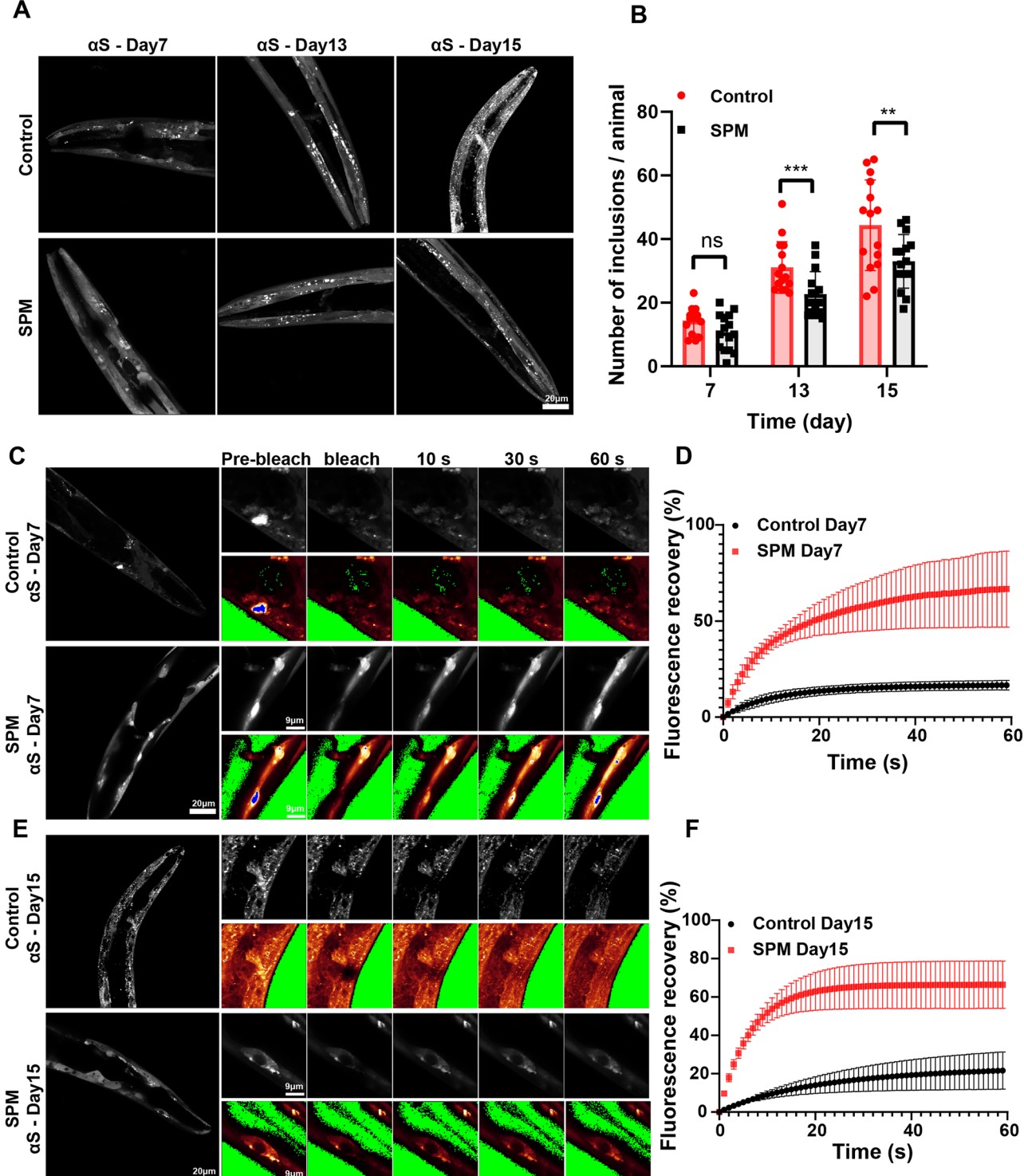

**Fig. 4 | Spermine decreases αS condensate formation and promotes mobility of the αS condensate. A** Representative αS-YFP images showing the effects of spermine in NL5901 worm strain. The top panel showing control administration and bottom panel showing spermine administration indicates the progression of the αS condensate assembly over time, the scale bar represents 20 μm. **B** Quantification of images shown in panel A of αS condensate in NL5901 worms at indicated time point in the absence and presence of spermine ($n = 15$) (day7: $p = 0.0522$, day13: $p = 0.0010$, day15: $p = 0.0090$). **C**, **D** Representative FRAP measurement images (**C**) and quantification of normalized recovery traces from FRAP

(**D**) of αS-YFP in the absence (top) and presence (bottom) of spermine at day 7. The images correspond to the region of interest with pre-bleach, post-bleach, 10 s, 30 s, and 60 s after bleach. The scale bar states in the images ($n = 3$). **E**, **F** Representative FRAP measurement images (**E**) and quantification of normalized recovery traces from FRAP (**F**) of αS-YFP in the absence (top) and presence (bottom) of spermine at day 15 ($n = 3$). The data represent the mean ± SEM. At least three worms were analyzed for each condition. A paired t-test was used in (**B**) (ns: not significant, *$P < 0.05$, **$P < 0.01$, ***$P < 0.001$).

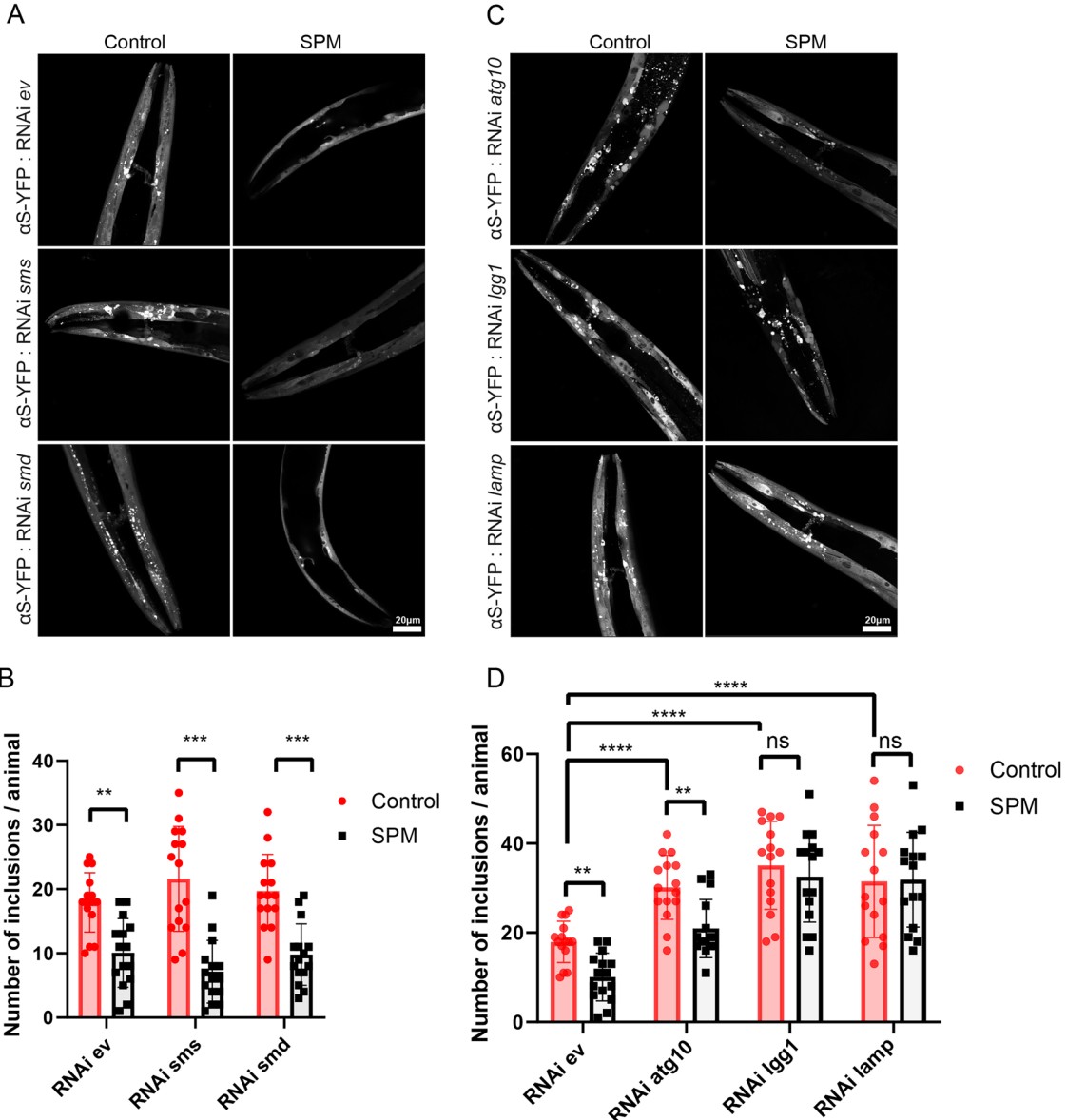

**Fig. 5 | Autophagy is essential for spermine-mediated αS degradation in *C.elegans* model of PD. A** Representative images of NL5901 worms treated with 500 μM of SPM (right) or vehicle control (left) and fed with *empty vector* (*ev*, top), *sms* RNAi (middle), *smd* RNAi (bottom). **B** Quantification of condensate of αS-YFP in NL5901 *C.elegans* that treated with *empty vector, sms, smd* RNAi, and SPM or vehicle control (*n* = 15). (RNAi *ev*: *p* = 0.0012, RNAi *sms*: *p* = 0.0001, RNAi *smd*: *p* = 0.0003. **C** Representative images of NL5901 worms treated with 500 μM of SPM (right) or vehicle control (left) and fed with *atg-10* RNAi (top), *lc3* RNAi

(middle), and *lamp* RNAi (bottom). **D** Quantification of condensate of αS-YFP in NL5901 *C.elegans* that treated with *empty vector, atg-10* RNAi, *lc3* RNAi, *lamp* RNAi, and SPM or vehicle control (*n* = 15). (RNAi *ev*: *p* = 0.0012, RNAi *atg10*: *p* = 0.0085, RNAi *lc3*: *p* = 0.5304, RNAi *lamp*: *p* = 0.9324, Control: RNAi *ev* vs RNAi *atg10*, *p* < 0.0001; RNAi *ev* vs RNAi *lc3*, *p* < 0.0001; RNAi *ev* vs RNAi *lamp*, *p* = 0.0005;)The scale bar is: 20 μm. The data represent the mean ± SEM. A paired t-test was used in (**C, D**) (ns: not significant, *$P < 0.05$, **$P < 0.01$, ***$P < 0.001$).

In addition to the spermine treatment, we further assessed the effect of endogenic spermine on αS condensates using the same *C. elegans* model NL5901. We fed these worms with different RNAi bacteria (empty vector, *sms* RNAi and *smd* RNAi) to silence the spermine synthase (*SMS*) and S-adenosylmethionine decarboxylase (*SMD*) genes, and verified by qPCR (Supplementary Fig. 22). Both *SMS* and *SMD* genes play critical roles in the spermine synthesis pathway[30] (Supplementary Fig. 23). The *SMS* gene encodes the enzyme spermine synthase, which catalyzes the synthesis of spermine from spermidine[31]. *SMD* is an enzyme that catalyzes the decarboxylation of S-adenosylmethionine (SAM) to produce decarboxylated S-adenosylmethionine (dcSAM). This reaction is a key step in the biosynthesis of polyamines[32]. The coordinated action of *SMS* and *SMD* is therefore integral to both the synthesis and degradation of

polyamines, maintaining cellular homeostasis, and affecting a wide array of physiological and developmental processes[33]. Figure 5A shows representative images for αS-YPF condensates in the presence and absence of RNAi and spermine. We observed a reduction in condensate formation following spermine treatment in the RNAi empty vector compared to the control (Fig. 5A, B), suggesting that spermine-induced condensates are degraded. The *sms* and *smd* RNAi groups exhibited enhanced αS condensate deposition, indicating that a deficiency in the spermine biosynthesis pathway contributes to αS accumulation. Additionally, we conducted further measurements with an external spermine supplement for all groups. In all cases, external spermine consistently reduced αS condensates (Fig. 5A, B), indicating that spermine, whether from endogenous or external sources, plays a crucial role in αS degradation in the *C. elegans* model.

To further elucidate the role of spermine in the degradation of these condensates, we investigated the autophagy degradation regulators. Autophagy is a cellular process that degrades and recycles toxic cytoplasmic components to maintain cellular homeostasis[34]. LGG-1, the *C. elegans* homolog of mammalian LC3 encoded by *lgg-1* gene, is a component of the autophagosome membrane, helping in expansion and completion of autophagosome[34,35]. The autophagosome fuses with a lysosome to form an autolysosome, a process facilitated by LAMP encoded by *lam-1*. LAMP is located on the lysosomal membrane and is essential for the fusion of the autophagosome with the lysosome. Within autolysosome, toxic proteins and other contents are degraded by lysosomal enzymes, effectively clearing them in cells[36,37]. We used RNAi to selectively silence target genes (*lc3* and *lamp*) and verified by qPCR (Supplementary Fig. 22), corresponding to different autophagy regulators. There is no significant difference in the total condensates observed administrative group with spermine or vehicle control when we knock down *beclin-1* or *lamp a*lthough spermine decreased the condensate in the empty vector group (Supplementary Fig. 24 & Fig. 5A). Interestingly, silencing *lgg-1* (LC3) modestly increased the spermine-mediated reduction of condensates (Fig. 5C, D), supporting the conclusion that spermine-induced clearance is autophagy regulator-dependent.

To further test the autophagy pathway, we examined other autophagy-related regulators, especially related to autophagosome formation. The *atg-10* gene plays a crucial role in the autophagy pathway by facilitating the conjugation of ATG12 to ATG5, an essential step in the early stages of autophagosome formation[14]. Importantly, there are the least condensates with spermine treatment in *atg-10* RNAi knockdown group in comparison to the vehicle control and other groups (Fig. 5C, D). Together with the observed effect of spermine under LC3 knockdown conditions, we propose that spermine facilitates the expansion of autophagosomes containing αS liquid-like condensates, thereby enhancing their degradation through the autophagy pathway. This is further supported by in vitro and ex vivo studies showing that spermine-mediated αS condensates at the mesoscopic level are dynamic and interact readily with larger components involved in the autophagy pathway.

### Spermine extends lifespan and improves fitness in AD and PD *C.elegans* model

Given the role of spermine in promoting phase separation and degradation of αS via autophagy, we further investigated the lifespan of the *C. elegans* models of AD and PD with the presence of spermine. Initially, we aimed to assess the therapeutic potential of spermine in neurodegenerative diseases by evaluating its effects on a nematode model of AD, encompassing strains VH255 that express wild-type Tau352 fragment[38] and BR5270 that has neuronal overexpression of human fragment of K18 ΔK280[39]. The K18 ΔK280 fragment has been shown to exhibit strong toxicity and fibrillation in vitro[40]. As expected, the VH255 and BR5270 control groups exhibited significantly shorter lifespan by 33.3% and 29.6% compared to N2 wild type controls, which indicated that Tau352 and K18 ΔK280 fragments both caused shorter lifespan. In contrast, treatment with spermine at the low concentration (100 μM) had no effect on lifespan (Supplementary Fig. 25A & Supplementary Fig. 26A). Thus, we decided to continue with a concentration-dependent treatment of spermine. We found a clear dose-response effect on lifespan with increasing spermine concentration from 100 to 500 μM. At 500 μM, spermine significantly delayed the mortality observed at advanced ages in both models (Supplementary Fig. 25A & Supplementary Fig. 26A). We next evaluated the effect of spermine on worm activity by assessing the bend frequency and head swing frequency at Day 6, 10 and 14, all of which were robustly improved by spermine in a dose-dependent manner during aging as compared with the vehicle (Supplementary Fig. 25B, C & Supplementary Fig. 26B, C).

Mitochondria participates in multiple metabolic pathway such as oxidative phosphorylation and TCA cycle[41]. Mitochondrial dysfunction leads to the accumulation of reactive oxygen species (ROS) and unbalance of calcium concentration, thereby accelerating aging and the progression of AD and PD[42]. It is well established that AD's Tau and PD's αS can lead to mitochondrial dysfunction (Fig. 6E, F). Therefore, we examined the effect of spermine supplementation on the mitochondrial function including ROS production and calcium concentration. As expected, the VH255 and BR5270 control groups exhibited significantly mitochondrial dysfunction including high ROS production detected by H2DCF-DA fluorescence probe and high calcium concentration compared to N2 WT worm (Supplementary Fig. 25D & Supplementary Fig. 26D). However, we observed a significant spermine concentration-dependent improvement in rescuing mitochondrial dysfunction, including a reduction in ROS production (Supplementary Fig. 25D & Supplementary Fig. 26D) and decreasing calcium concentration (Supplementary Fig. 25E & Supplementary Fig. 26E) by supplementation of spermine to VH255 and BR5270 Tau *C. elegans* models.

To evaluate the potential of spermine as a treatment for PD, we tested the impact on a nematode model NL5901 of PD. Similar to AD models, NL5901 control group exhibited significant shorter lifespan by 25.9% compared to N2, which is because overexpression of αS caused strong damage on lifespan. However, it showed a dose-response effect on lifespan after the spermine treatment. 500 μM spermine significantly prolongs the lifespan at advanced ages in PD models (Fig. 6B). Then, we measured the locomotive ability on Days 6, 10, and 14. We found the significantly improved bend frequency (Fig. 6C) and heading swing frequency (Fig. 6D) on Day 14. Under the same conditions, spermine decreased the ROS production (Fig. 6E) and calcium concentration (Fig. 6F) of NL5901 of PD models on Day 14.

## Discussion

While much research effort has been dedicated to investigating Tau and αS LLPS[22], aggregation kinetics[43], fibril structure[44], templated misfolding, and pathological toxicity in diseased models[45], it is crucial to have a comprehensive understanding of the biochemical aspects of various proteins associated with LLPS and neurological disorders, particularly in the context of IDPs (intrinsically disordered proteins). Structural and functional differences among proteins with distinct primary sequences may govern the biophysical properties of phase-separated droplets within the intracellular conditions, subsequently influencing the fate and function of the protein in living cells.

Both Tau and αS have been previously shown to phase separate or aggregate in vitro[46,47]. In the case of αS, it has been reported that both phase separation and aggregation are mediated by electrostatic interactions of the positively charged N-terminal domain and CTT[47]. Though the role of charged amino acids in Tau LLPS is less clear, hyperphosphorylation or aggregation-prone mutations, which increase the β-sheet propensity in the microtubule-binding repeat domain of Tau and lead to Tau oligomerization and aggregation, alter the protein charge and conformation; these changes appear to be crucial for phase separation into droplets[48,49]. Spermine, a positively charged polyamine, is therefore capable of affecting both amyloid protein fibrillation[26,50–53] and droplet formation[54].

In this study, we demonstrate that spermine promotes phase separation of both Tau and αS, leading to the formation of liquid droplets with weak intermolecular interactions. Both Tau and αS droplets exhibited dynamic liquid-like properties in vitro, as confirmed by FRAP measurements. More notably, both endogenous and external spermine reduced αS condensation in a *C. elegans* model. The presence of spermine significantly altered the dynamic properties of the condensates, shifting them from a solid-like state with limited fluorescence recovery to a more dynamic, liquid-like state with rapid recovery. These findings highlight the critical role of spermine in modulating the dynamics of condensates in vivo.

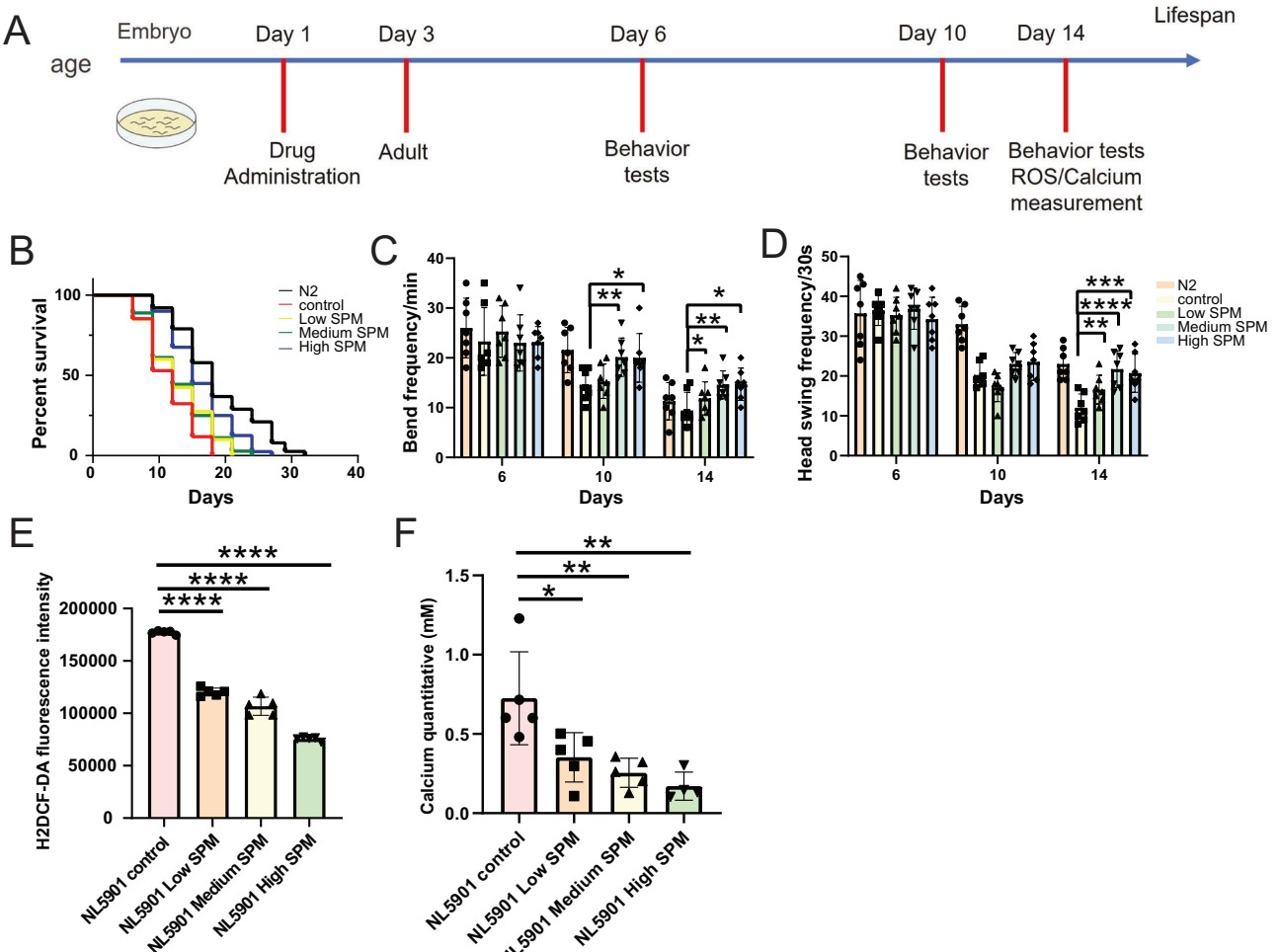

**Fig. 6 | Spermine improves lifespan and rescues movement deficits and mitochondrial dysfunction in *C.elegans* model of PD. A** Schematic overview of *C.elegans* experiments. **B** The lifespan of PD *C.elegans* model of NL5901 treated with control (OP50 vehicle), low (100 μM), medium (200 μM) and high concentration (500 μM) of spermine, which shows that spermine improves the lifespan of PD *C.elegans* model in a concentration dependent manner. **C, D** Spermine rescues movement deficits in PD *C.elegans* model. **C** bend frequency. (Day10: High SPM vs control, *p* = 0.0488, Medium SPM vs control, *p* = 0.0072; Day14: High SPM vs control, *p* = 0.0189, Medium SPM vs control, *p* = 0.0075, Low SPM vs control, *p* = 0.0467). **D** Head-swing frequency (*n* = 7), (Day14: High SPM vs control, *p* = 0.0008, Medium SPM vs control, *p* < 0.0001, Low SPM vs control, *p* = 0.0068). **E, F** Spermine rescues mitochondrial dysfunction in PD *C.elegans* model of NL5901. Reactive oxygen species (ROS, **E**), (*p* < 0.0001). Calcium concentration of mitochondrial (**F**) (*n* = 5). (High SPM vs control, *p* = 0.0087, Medium SPM vs control, *p* = 0.0091, Low SPM vs control, *p* = 0.0365). A paired t-test was used. Data are presented as mean values +/− SEM.

We next investigated the molecular mechanisms underlying spermine-induced phase separation of Tau and αS, focusing on their distinct charge distributions within the sequences. To explore this, we employed a range of biophysical techniques, including TR-SAXS, NMR, and coarse-grained molecular dynamics simulations. Our TR-SAXS results revealed that although both Tau and αS expand at later stages of spermine mixing, primarily driven by favorable intermolecular interactions, they exhibit different behaviors at the initial millisecond stage. Specifically, while αS consistently expands, Tau first undergoes collapse before expanding further, likely due to spermine-induced changes in intramolecular interactions. Our NMR experiments further confirmed that the uneven charge distribution leads to less specific interactions. Our insights into αS rely in part on prior NMR data[26], which indicated predominant specific interactions between the CTT of αS and spermine. In contrast, our NMR data on Tau showed a more uniformly distributed residue-level interaction pattern. Additionally, coarse-grained molecular dynamics simulations indicated that, although both αS and Tau exhibit spermine-induced expansion, the molecular mechanisms driving

their LLPS differ. While spermine-induced LLPS in αS can largely be explained by specific CTT-spermine interactions, spermine primarily neutralizes negatively charged amino acids throughout Tau and generally perturbs nonspecific interactions within the chain.

Despite their distinct charge distributions, likely related to their specific functions, both Tau and αS undergo phase separation upon spermine induction, suggesting potential biological roles in vivo. These condensates may prevent aggregation while serving as reservoirs in the presence of polyamines like spermine. In C. elegans models overexpressing Tau352, K18 ΔK280 Tau, and full-length human αS in AD and PD models, spermine treatment prolongs lifespan and rescues movement deficits and mitochondrial dysfunction caused by amyloidogenic proteins, likely through the function of amyloid-spermine condensate reservoirs. Reports of droplet-like Tau accumulation in neurons without aggregation suggest a physiological role for Tau LLPS[55], while our observations as well as other studies[56] of liquid-like αS droplets in PD *C.elegans* models, further indicate a distinction between LLPS and aggregation. Spermine promotes the mobility of condensates in the PD C. elegans model and delays the transition from

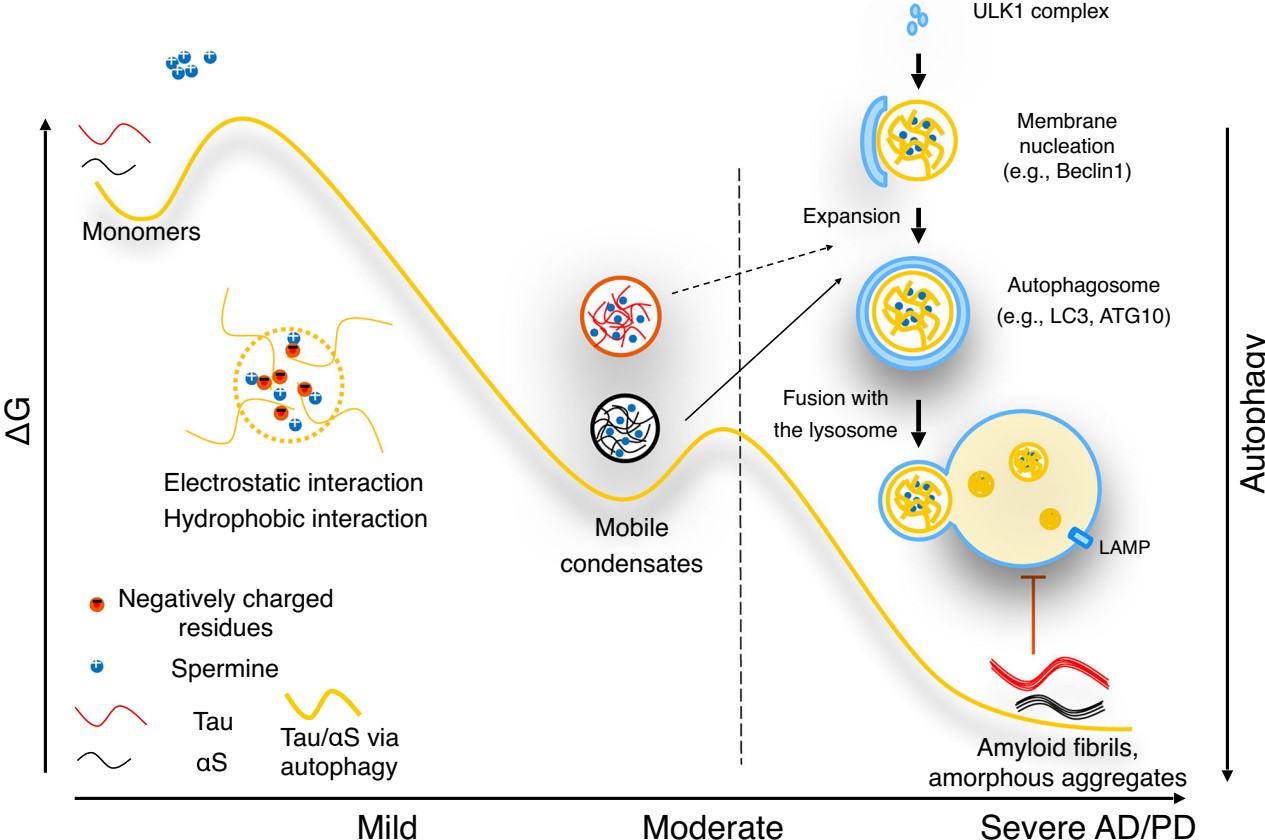

**Fig. 7 | Schematic representation of the proposed mechanism for amyloid LLPS and degradation.** Based on our observation, we propose a model for mobile amyloid condensate degradation via autophagosome expansion, which is initiated through spermine-induced LLPS. Under physiological conditions, Tau and αS do not undergo phase separation but form less dynamic solid-like condensates. In contrast, in the presence of spermine, Tau and αS both form dynamic liquid-like condensates with high mobility. These protein-dense condensates are interconnected by weak intermolecular interactions, making them more readily recognized and fused by the autophagosome and finally degraded via the lysosome.

liquid to solid state, providing the living cell with more opportunities to be recognized by the autophagy process compared to aggregation. While both LLPS and aggregation involve the assembly of biomolecules, they differ in their outcomes and underlying mechanisms. Based on our LLPS and ThT observation, spermine promotes Tau and αS LLPS but inhibits their fibrillation under the crowding conditions, highlighting its dual therapeutic potential in AD and PD treatment.

We observed that autophagy regulators ATG10, LAMP, and LC3 are involved in the spermine-induced αS condensate degradation. While our findings indicate that spermine-driven αS condensate clearance is primarily autophagy-dependent, we cannot exclude potential contributions from the ubiquitin–proteasome system. Further studies will be needed to determine whether spermine affects proteasome activity or αS ubiquitination, which could act in parallel or in coordination with autophagy to regulate αS turnover. We anticipate that mobilized condensates are readily recognized and fused with autophagy-related liquid autophagosomes based on the principle that "like dissolves like." Our data are consistent with the hypothesis that liquid-like condensates are more accessible to autophagic clearance than fibrillar assemblies, although they do not provide conclusive proof. Interestingly, in the spermine-treated groups, RNAi knockdown of *atg10* and *lc3* resulted in more reduction in αS inclusion numbers, whereas knockdown of *beclin1* and *lamp* did not show comparable effects. This discrepancy may reflect the stage-specific roles of these autophagy components. ATG10 and LC3 proteins function at an intermediate step in autophagosome biogenesis by facilitating phagophore membrane expansion, whereas LAMP are involved in later

steps—autophagosome maturation and lysosome fusion, respectively. We propose that spermine stabilizes αS condensates in a highly mobile and dynamic state, thereby promoting interactions with early autophagy machinery such as the ATG10 protein-associated complex. In contrast, the downstream components (LAMP) may reduce accessibility or engagement with these dynamic condensates during the early clearance phase. This stage-selective interaction may explain the stronger effect observed upon *atg10/lc3* knockdown and underscores the importance of early autophagic events in spermine-facilitated clearance of αS condensates. Multiple studies documented the in vivo formation of p62/SQSTM1 condensates, and in vitro experiments revealed phase separation when reconstituted p62/SQSTM1 was mixed with polyubiquitin chains[57,58]. These cohesive bodies exhibited a semi-liquid nature, further emphasizing the role of p62/SQSTM1 in cellular processes. Intriguingly, LC3B was identified as a negative regulator of p62/SQSTM1 phase separation[36]. p62 also functions as an autophagy receptor, recognizing depolarized aggregates through ubiquitin chains. Subsequently, p62 isolates and transfers damaged protein by interacting with LC3, contributing to the cellular quality control mechanisms[14]. We anticipate that mobilized condensates with spermine is recognized and fused by p62 condensates to involve in autophagy process. However, further investigations are needed to explore the role of p62 in this context, which lies beyond the scope of our current study.

Based on our findings, we propose a model in Fig. 7 and Table S1 that illustrates the function and structure of Tau and αS condensates modulated by spermine, along with a mechanism by which these

condensates contribute to cellular protein homeostasis through degradation pathways. Spermine appears to play a central stabilizing role in maintaining condensates in a liquid-like state at elevated energy levels for an extended period. This property allows subcellular systems to recognize and respond to condensates before they transition into a solid, lower-energy state. Our model suggests a dynamic interaction with cellular autophagy processes, where liquid-like condensates are recognized and cleared by autophagosomes. However, once these condensates mature into solid-like fibrils, they become resistant to autophagic fusion and clearance, contributing to the pathology seen in neurodegenerative diseases such as Alzheimer's and Parkinson's. The ability of autophagosomes to target early-stage fibrils suggests a critical window that could prevent the progression of protein aggregation diseases. Overall, our data underscore the significance of spermine-induced condensates and their potential degradation via autophagy. Understanding how spermine modulates the intermolecular interactions of amyloidogenic proteins offers a strategy for developing therapeutic molecular glues aiming at preventing or degrading age-related aggregates.

## Methods

### Protein sample preparation

Recombinant full-length human Tau (441 amino acid, UniProt P10636-8), the K18 variant of Tau (residues 244 to 372) retaining core aggregation-prone sequence[59], and human αS (140 amino acid, UniProt P37840-1) were expressed in *E.Coli* BL21 (DE3) and purified as described previously[60,61]. Briefly, the expression of Tau, K18 and αS proteins was conducted at 37 °C in LB medium containing ampicillin (100 mg/L) and initiated by adding a final concentration of 0.4 mM Isopropyl β-d-1-thiogalactopyranoside (IPTG) to the bacterial cultures to an OD600 of 0.6-0.8 and continued culture at 37 °C for 2 h for Tau and K18 expression and at 20 °C for 24 h for αS expression. The cells were harvested, and the pellets were stored at −20 °C before further purification.

The Tau and K18 pellets were resuspended in 50 mM NaPi, 2.5 mM EDTA, pH 6.2, supplemented with a tablet of protease inhibitor cocktail. The soluble extracts were obtained by sonication step for 3 min (5 s on, 5 s off, 30% amplitude) and centrifugation at 25,000 g for 30 min. The supernatants were heated at 75 °C for 15 min, and the insoluble materials were removed by centrifugation for 20 min at 25,000 g. The supernatant was loaded on a Hi-Trap SP FF cation exchange chromatography column (GE healthcare), and the proteins were eluted by a NaCl gradient using 50 mM NaPi, 2.5 mM EDTA, 500 mM NaCl pH 6.2, and fractions were collected. To label K18 and Tau with $^{15}N$ stable isotopes, the *E. coli* culture expressing K18 and Tau protein was grown in a M9 minimal medium with $^{15}NH_4Cl$. The purification process was the same as unlabeled Tau.

The αS pellets were resuspended in 25 mM Tris-HCl, pH 8.0, and subsequently precipitated using ammonium sulphate after a lysis step by sonication and heating at 75 °C for 15 min. The supernatant after centrifugation was loaded onto a Hi-Trap Q HP anion exchange chromatography column. The protein was eluted using a gradient of NaCl with 25 mM Tris-HCl, 800 mM NaCl pH 8.0. The eluted αS fractions were combined and precipitated with ammonium sulphate. Subsequently, the pellet was resuspended, and size exclusion chromatography was performed using a Superdex 200 Increase 10/300 GL column to obtain a monomeric fraction of αS.

The purified proteins, checked by 12% SDS-PAGE, were dialyzed against 20 mM ammonium bicarbonate buffer prior to lyophilization. Protein concentration was determined by measuring UV-absorption at 280 nm (extinction coefficient of 5600 $M^{-1} cm^{-1}$ of αS, 7600 $M^{-1} cm^{-1}$ of Tau, 1490 $M^{-1} cm^{-1}$ of K18).

### Protein labeling

For Tau protein fluorescence labelling, Alexa Fluor™ 647 C2 Maleimide was used as per the manufacturer's instruction (Catalog No. A20347,

ThermoFisher Scientific, USA). In brief, Dissolve the lyophilized Tau protein to 100 μM in 50 mM phosphate buffer at pH 7.4 on ice. 1 mM of a reducing agent DTT was dropwise to protein solution, followed by incubation at 4 °C for 30 minutes. Excess DTT was subsequently removed using a spin desalting column (Catalog No. 89882, ThermoFisher Scientific, USA). Prior to labeling, a 10 mM stock solution of Alexa Fluor™ 647 C2 Maleimide was prepared in dimethyl sulfoxide (DMSO). A 20-fold molar excess of the dye reagent was added dropwise to the protein solution under constant stirring, and the reaction was allowed to proceed overnight at 4 °C. Unreacted dye was removed using a spin desalting column equilibrated with 50 mM phosphate buffer. To prevent photobleaching, all dye solutions and labeled protein samples were protected from light by wrapping containers in aluminum foil.

### αS A140C mutant labelling

αS A140C mutant protein was purified following the same protocol with αS wild type protein purification mention before. For αS A140C labelling, Alexa Fluor™ 647 C2 Maleimide was used. The αS A140C protein was mixed with the 20-fold molar excess of dye for overnight at 4 °C. The excess dye was removed by using the spin desalting column in 50 mM phosphate buffer pH 7.4. For further experiments, we used 5% ratio of labeled versus unlabeled protein, unless mentioned otherwise.

### Measurements of liquid-liquid phase separation in vitro

All protein samples were prepared on ice. For each tested condition, protein and molecular crowding agent polyethelene glycol 8000 (PEG-8000, Sigma, CAS: 25322-68-3) was dissolved using 25 mM HEPES buffer (pH 7.4). All buffer solutions were degassed. Stock protein samples were centrifuged for 20 min at 15,000 g to remove aggregated protein. Stock spermine was dissolved with PEG in 25 mM HEPES buffer. The LLPS experiments were performed in LCP crystallization plate. The samples were dispensed by using the multi-channel pipetting robot mosquito (SPT Labtech). 200 nL protein was mixed 1:1 with the spermine-PEG solution to achieve the desired concentration. At least 3 replicates per condition were tested. The plates were sealed and kept at 20 °C and imaged at a certain time point using a Rock Imager microscope (Formulatrix). The droplet size was calculated using the Fiji image analysis platform[62].

### Fluorescence imaging and FRAP analysis in vitro

The fluorophore-labeled LLPS were conducted using Stellaris microscope (Leica Microsystems, Germany). Droplet formation was using 5% labeled protein + 95% unlabeled protein. The sample prepared followed the same previously description as the unlabeled system. The LCP plate was taken out from the Rock imager system and mounted to the Stellaris microscope. Imaging was acquired using the 647 nm laser line and a 63× oil-immersion lens. All images were acquired using identical laser power and detector gain settings; however, images from different wells may account for the slight differences in background intensity observed.

Fluorescence recovery after photobleaching (FRAP) experiments were performed using the same microscope with FRAP mode. For each droplet, the selected ROI (region of interest) was bleached at 60% laser intensity for 2 s, and post-bleach time-lapse images were collected (1 s frame rate, 60 frames) and analyzed with plugins FRAP_profiler_v2 developed by the Hardin lab in Fiji[63]. For FRAP analysis, the fluorescence intensities of the two regions were recorded (ROI1 = photobleached region, ROI2 = unbleached region to correct the background signal over time). The recovery time constant was derived from a single exponential fit of the corrected fluorescence intensities.

### Time-resolved Small Angle X-ray scattering (TR-SAXS) measurement

To determine changes in single-chain dimensions of Tau and αS protein in the two-phase regime triggered by spermine, we used time-

resolved small-angle X-ray scattering (TR-SAXS) experiments coupled with an adaptive microfluidic chip (Microfluidic ChipShop, Germany, Product Code: 10000258) with three channels. All three channels are syringe pump driven, where the middle channel was applied for protein solution and both side channels were used to pump buffer or spermine solution, schematically shown in Fig. 2.

Synchrotron SAXS data ($I(q)$ vs $q$, where $q=4\pi sin\theta/\lambda$, is the scattering vector, and $\lambda = 0.124$ nm is the wavelength) were measured on the CoSAXS beamline at MAX IV, Lund, Sweden, equipped with a Eiger2 4 M SAXS detector and Pilatus 2 M WAXS detector. Scattering data were acquired at room temperature for Tau and αS in the absence or presence of spermine (100 μM of protein, the ratio of 1:10). The scattering data were recorded as 30×30 ms frames for 50 repetitions. Subtraction and data processing were performed using the ATSAS software package including PRIMUS for the evaluation of the radius of gyration ($Rg$), and extrapolated forward scattering at zero-angle, $I(0)$, from the Guinier approximation (for $qRg < 1.3$)[64]. Ensemble optimization method was applied to fit the raw TR-SAXS data using the EOM program version 2.1[25].

## NMR measurements
To study the interaction of $^{15}$N-Tau and $^{15}$N-K18 with spermine, two-dimensional $^{1}$H-$^{15}$N HMQC spectra were recorded on a Bruker Avance III 700 MHz spectrometer, equipped with a triple resonance cryogenic probe, at 283 K either for the protein (100 μM protein in 20 mM sodium phosphate, pH 6.3, 10% D2O) alone or using a 1:10 protein: spermine molar ratio. For HMQC experiments, the number of scans was 19,0 and the TD for the F2 and F1 dimensions were 2048 and 128 points, respectively. The spectral width for the F2 and F1 dimensions was 14.2830 and 40.2688 ppm, respectively. The acquisition times for the F2 and F1 dimensions were 0.1024000 and 0.0224000 seconds, respectively.

The spectra were processed using Bruker Topspin. Data was analyzed using the software's Topspin v.4.2.0 and Poky. The assignments (BMRB entry 50701 and 19253 for Tau and K18, respectively) used to assign and analyze the data have been published previously by other groups[65,66].

Normalized weighted average chemical shift changes $\Delta\delta$ (Eq. 1) for amide $^{1}$H and $^{15}$N chemical shifts upon the addition of spermine onto Tau were determined[67].

$$\Delta\delta = (((\Delta\delta_{N}/5)^2 + (\Delta\delta_{H})^2)/2)^{1/2} \qquad (1)$$

## ThT kinetics
To evaluate the effect of spermine on the aggregation kinetics of Tau and αS, 15 μM Tau and 50 μM αS were incubated with varying molar ratios of spermine in the presence of 10% PEG8000, 20 μM Thioflavin T (ThT), and two 1.0 mm diameter glass beads (Sigma-Aldrich). All samples were prepared on ice, and 35 μL aliquots were dispensed into a 384-well black/clear-bottom microplate (ThermoFisher, cat. no. 242764) and sealed to prevent evaporation. Aggregation kinetics were monitored using a PHERAstar FSX microplate reader (BMG LABTECH, Germany), with ThT fluorescence measured every 10 min (excitation: 430 nm; emission: 480 nm). Experiments were performed at 37 °C with continuous shaking at 300 rpm in 25 mM HEPES buffer (pH 7.4), using five technical replicates per condition.

## Mass photometry
Mass photometry was conducted using the OneMP mass photometer (Refeyn, UK). Rectangular glass slides and self-adhesive silicone wells were cleaned following the manufacturer's protocol. The glass slides were mounted onto a 100X oil immersion objective, and the silicone well was affixed to facilitate sample loading. For calibration, β-Amylase

and Thyroglobulin served as standard proteins. Calibration was conducted using a laser for 120 seconds under optimized auto-exposure settings.

Tau protein was prepared in 25 mM HEPES buffer (pH 7.4) to a total volume of 50 μL in a 1.5 mL Eppendorf tube. For mass photometry measurements, 18 μL of the buffer was loaded onto the silicone well, followed by autofocusing. Subsequently, 2 μL of sample was added to achieve a final concentration of 10 nM, and mass measurements were performed with the laser for 60 seconds. Data analysis was conducted using DiscoverMP software. Molecular weight estimations were based on the contrast values of individual landing events captured across multiple frames, generating a contrast-to-molecular weight (MW) calibration curve. Histograms of detected particles were fitted with Gaussian functions to determine the average MW of the samples.

## Coarse-grained molecular dynamics simulations
A previously developed residue-based coarse-grained model, HPS model[27], is used to study the liquid-liquid phase separation of Tau and αS. Each amino acid is represented by one bead with its charge and hydropathy. In addition to the three types of interactions (bonded interactions, electrostatic interactions, and short-range pairwise interactions) originally introduced in the HPS model, we further added additional terms for angle[68] and dihedral potentials[69], similar to the previous work for studying LLPS of αS[20]. The HOOMD-Blue software v2.9.3[70] together with the azplugins (https://github.com/mphowardlab/azplugins) was used for running the molecular dynamics simulations. For the simulations with single chains, one chain of Tau or αS, has been simulated without spermine and with 20 spermine molecules. For the LLPS simulations, we have inserted 96 chains of protein in the simulation box without spermine and with spermine at 1:1, 1:3, 1:5, 1:10, and 1:15 ratio. All simulations were run using a Langevin thermostat with a friction coefficient of 0.01 ps$^{-1}$, a time step of 10 fs and a temperature of 298 K. For single-chain simulations, the simulations were run for 5 μs and the first 0.5μs were dumped for equilibration. For co-existence simulation using a slab initial conformation, the simulations were run for 5 μs, and the first 1 μs were dumped for equilibration. The analysis was done using MDAnalysis[71].

## C. elegans strains and maintenance
Standard procedures were used for the propagation of *C.elegans*. All strains were cultured at 22 °C on nematode growth agar medium plates (NGM: 1 mM CaCl$_2$, 1 mM MgSO$_4$, 5 lg/ml cholesterol, 250 mM KH$_2$PO$_4$, 17 g/L agar, 3 g/L NaCl, and 7.5 g/L casein,), which were seeded with the *E.coli* strain OP50. Worms were synchronized by hypochlorite bleaching, washed in M9 buffer (3 g/L KH$_2$PO$_4$, 6 g/L, Na2HPO4, 5 g/L NaCl, and 1 mM MgSO$_4$), and subsequently transferred to a seeded NGM plate to culture.

The following strains were used in this study: N2 Bristol (wild type). NL5901: pkIs2386 [unc-54p::α-synuclein::YFP + unc-119( + )], BR5270: byIs161 [rab-3p::F3(delta)K280 + myo-2p::mCherry], VH255: hdEx82 [F25B3.3::Tau352(WT) + pha-1(+)].

## Lifespan assay
Lifespan analyses were performed at 22 °C. Age-synchronous animal populations were generated by bleaching (hypochlorite treatment) gravid adult worms of the desired strain. Eggs were placed on NGM plates seeded with OP50 until the L4 stage. Then animals were transferred to fresh NGM-FUDR plate treated with different concentrations of spermine. The dead worms were counted every 2 days until all nematodes died by determining their touch-provoked movement and pharyngeal pumping. Worms that desiccated due to crawling on the edge of the plates were censored and incorporated as such into the dataset. Approximately 40 worms were scored in each experiment, 3 independent assays per conditions were measured.

## Behavior assay

To investigate how the treatment with spermine improves the behavioral ability, adult nematodes after day 6, day 10 and day 14. Behavior frequency was determined by picking 7 worms from the bacterial lawn on an NGM agar plate and transferring them onto a bacteria-free plate, because of the difficulty to observe behavior on food-containing agar plates as worms tend to hide under the bacteria. Heading swing frequency and bend frequency were observed for each of the 7 worms for 1 min at room temperature using a dissection microscope (Leica).

## Measurement of mitochondrial function

Reactive Oxygen Species (ROS) production: the assays were conducted using a ROS detection kit (abcam, ab113851). Endogenous ROS levels were measured using 2′, 7′-dichlorofluorescein diacetate (H2DCF-DA). At 14 days, one hundred worms were collected and washed twice with PBS and once with PBS supplemented with 1% Tween20. The worms were resuspended with PBS before sonication. The worms were sonicated on ice for two times for 15 s (1 s on, 3 s off, 30% amplitude) using a small-probe sonication. H2DCF-DA was added to a final concentration of 10 μM and incubated for 30 min at 37 °C. Fluorescence intensity (excitation and emission wavelengths of 485 and 520 nm, respectively) was measured with a PHERAstar FSX microplate reader (BMG LABTECH, Germany). The assay was performed three independent times.

Calcium concentration: the assays were conducted using a calcium quantification kit (abcam, ab112115). The worm suspensions were prepared in the same way as described for the ROS measurement. After sonication, the solution was mixed 1:1 with RhodRed (200x diluted) and incubated at room temperature for 20 min in a microplate. Fluorescence intensity (excitation and emission wavelengths of 575 and 620 nm, respectively) was read with a PHERAstar FSX microplate reader (BMG LABTECH, Germany). The assay was performed three independent times.

## Fluorescence imaging and FRAP analysis of *C.elegans*

For fluorescence microscopy, worms were mounted on 2% agarose pads and immobilized in 20 mM sodium azide. Fluorescence microscopy and FRAP analysis were performed on a Leica Stellaris confocal microscope with 63X objective lens. FRAP was carried out by using the 63x objective lens at 5x zoom, with the 514 nm laser for excitation of YFP and 80% power intensity. The FRAP analysis followed the same previously description as in vitro.

## RNAi experiments in *C.elegans*

RNAi feeding protocol were used for RNAi inactivation[72]. Bacterial clones expressing double-stranded RNA were obtained from the Ahringer RNAi library. Individual RNAi clones were streaked from the desired library wells onto LB agar plates supplemented with 50 μg/mL ampicillin and incubated overnight. Single colonies were inoculated into 1 mL of LB medium containing 40 μg/mL ampicillin and grown overnight at 37 °C with shaking. Starved C. elegans plates were chunked onto fresh NGM plates seeded with E. coli OP50 and allowed to recover for two days. For RNAi feeding, 200 μL of each bacterial culture was spotted onto NGM agar plates containing 1 mM IPTG and 25 μg/mL carbenicillin and spread evenly over the surface. Gravid hermaphrodites were treated with alkaline hypochlorite solution to release eggs, which were washed in M9 buffer and subsequently transferred to the RNAi plates seeded with E. coli HT115(DE3) expressing the corresponding double-stranded RNA.

## Quantitative real-time PCR (qPCR)

qPCR was performed to assess gene expression in *C. elegans*. Approximately 1000 worms per sample were collected from NGM plate using 15 mL M9 buffer and pelleted by centrifugation at 4000 rpm for 1 min. This washing step was repeated three times. The worm pellet was resuspended in 1 mL TRIzol™ Reagent (Thermo Fisher Scientific) and subjected to three freeze-thaw cycles alternating between liquid nitrogen and room temperature to ensure complete lysis. Lysates were incubated at room temperature for 5 min, followed by the addition of 200 μL chloroform. Samples were incubated on ice for 3 min and centrifuged at 12,000 rpm for 15 min at 4 °C.

Approximately 450 μL of the upper aqueous phase was transferred to an RNase-free tube and mixed with an equal volume of isopropanol to precipitate RNA. The mixture was centrifuged at 12,000 rpm for 10 min at 4 °C, and the RNA pellet was washed sequentially with 1 mL of 75% ethanol and 1 mL of anhydrous ethanol. The pellet was air-dried and dissolved in 20 μL RNase-free water. RNA concentration and purity were assessed using a NanoDrop, and samples were stored at −80 °C.

qPCR reactions were performed using SYBR Select Master Mix (Bio-Rad) in a final volume of 20 μL containing 10 μL of 2× Supermix, 0.5 μL of each primer (250 nM final concentration), 2 μL of 1:10 diluted cDNA, and nuclease-free water. Amplification was conducted on a Bio-Rad CFX96 Real-Time PCR Detection System following the manufacturer's cycling protocol. Melt curve analysis was performed at the end of each run to verify amplification specificity. Relative gene expression levels were calculated using the ΔΔCt method and normalized to β-actin expression. Primer sequences used for qPCR are listed in Table S2.

## Quantification and statistical analysis

All statistical analysis was performed in GraphPad Prism or Origin. Data are presented as mean ± SEM (standard error of the mean) from at least three independent biological replicates, unless indicated otherwise. Statistical significance between experimental groups was analyzed either by two-tailed Student's t-test or one-way ANOVA followed by Bonferroni's multiple comparison.

## Reporting summary

Further information on research design is available in the Nature Portfolio Reporting Summary linked to this article.

# Data availability

All data are available in the main text or the Supplementary information. Source data are provided with this paper.

# Code availability

The script for performing the coarse-grained molecular dynamics simulation is shared in Zenodo [https://doi.org/10.5281/zenodo.17179918].

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

## Acknowledgements

The authors acknowledge the support from the Swiss National Scientific Foundation (310030_197626 & 10002967, J.L.), the Brightfocus Foundation (A20201759S, J.L.), the National Institutes of Health (R35GM146814, W.Z.), the China Scholarship Council (202004910346 to X.S.) and the research computing facility at Arizona State University (W.Z.). We thank Prof. Anne Spang and Prof. Jan Pieter Abrahams (University of Basel) for their insightful comments and helpful discussions. We also gratefully acknowledge beamtime and support from the ESRF (ID02 beamline, Grenoble), EMBL Hamburg (P12 BioSAXS beamline), MAX IV Laboratory (CoSAXS beamline), and the Swiss Light Source (cSAXS beamline) for enabling and supporting our SAXS experiments.

## Author contributions

X.S., D.S., W.Z. and J.H.L. conceptualized the study. X.S., D.S., X.W., C.M., R.S.H., J.A.G. and F.H. performed the research and developed the methodology. D.S., W.Z. performed MD simulations and analyzed the data. X.S., X.W. performed the *C. elegans* experiments. X.S., C.M., J.A.G.; R.R. performed NMR data collection and data analysis. J.H.L. supervised all aspects of the study. The original manuscript draft was written by X.S. and J.H.L., with review and editing by all authors.

## Competing interests

The authors declare no competing interests.
