## [Transparent Peer Review file · Nature Communications]

Spermine Modulation of Alzheimer's Tau and Parkinson's α -Synuclein: Implications for Biomolecular Condensation and Neurodegeneration

Corresponding Author: Dr Jinghui Luo

Version 0:

Reviewer comments:

Reviewer #1

(Remarks to the Author)

In this interesting research project, the authors discussed the effects of spermine on the conformation, interaction and LLPS of Alzheimer's Tau and Parkinson's alpha-Synuclein. The research design is consistent including utilizing the advanced experimental methods and simulations, and the results provide new insights for biomolecular condensation and neurodegeneration. My comments focus on the phase behavior (LLPS) and the TR-SAXS measurements and results. Specific comments are listed below;

1. Line 134-135, Fig.1F, the authors claim "In the presence of 1,6-hexanediol, Tau droplets without spermine were completely dissolved, while Tau droplets with spermine remained intact (Fig.1F)." The statement for Tau droplets is not fully correct. As one can see in Fig.1F, with increasing spermine concentration, the number of droplets is clearly reduced. When further increasing spermine concentration, Tau droplets may also dissolve.
2. Fig.1, the effect of adding spermine on the LLPS of Tau and alpha-S shows some unconventional results. First of all, the phase diagram in Fig.1C and 1I clearly indicates that with increasing SPM the LLPS region is expanded, that is as the authors claimed "SPM enhance LLPS". However, the Fluorescence results in Fig.1B and H also clearly indicates that the protein concentration difference in the droplets (dense phase) and the dilute phase in fact are reduced with increasing SPM concentration. These observations seem contradictory to each other.
3. Line 213-214 "The Tau concentration was increased from 0 to 100 μ M..." is this true? In the methods section (line777), it states that the protein concentration is fixed to 100 μ M, with and without SPM.
4. For TR-SAXS measurements and results in Fig.2. Fig.S9,S10. One suggestion is that in Fig.S9 and S10, maybe better using double log-scale to emphasize the low q region as both Guinier analysis and the I0 information mainly from this region. Currently, the Intensity was plotted in log-scale but the label "Intensity" maybe a mistake.
 - a. Line 222-223, "measurement of Tau with spermine showed that the compaction of Tau with a decay constant of ~ 0.2 s". It is not clear to me what this decay constant means and how this value is determined. Furthermore, in Line224-225, "the time constant of tau collapse was of the same order as the initial stage". Again, not sure how the initial stage is defined.
 - b. Line 231-233 "the shapes of the two SAXS profiles overlaid well when normalized by I0 and Rg,...". First of all, in Fig.S10, the plots were not normalized. Secondly, if the statement here is correct, that the "reduction in size is from the decrease in average dimension, rather than complete or partially folding", Isn't this controversy to the observed increase of Rg?
 - c. Increase of Rg for Tau over time, may indicate the aggregation instead of expansion. The current TR-SAXS measurements cannot exclude this option.
 - d. From Line 208 to line 251, the description in these two paragraphs has some sentences repeated, such as "Experimental TR-SAXS data, depicted in red,...". More concise is needed.
 - e. The conclusions in this section require more support or discussion. "Tau underwent rapid initial compaction followed by an increase in its Rg, with the initial stage indicating a barrier-limited collapse driven by intramolecular interactions, while the subsequent expansion may be driven by more favorable intermolecular interactions." In this statement, a barrier-limited collapse needs to be proved, and the intramolecular interactions need to be discussed. For the subsequent increase of Rg, on the one hand, as indicated above, increase of Rg may be due to protein aggregation. The authors need to demonstrate this is not this case. It is also not clear why "more favorable intermolecular interactions" would drive the expansion instead of aggregation. One would expect an intramolecular repulsion to cause the expansion.

5. Fig.3C, Line 330-333, not clear how to reach this conclusion “whereas for Tau, Spermine helps in overall expansion of the chain which explains the increases in Rg.” Further, the authors concluded that “Spermine can be considered as a surfactant to coat the negatively charged regions of the chain and reduces attractive interactions ...” if this is true, then it is conflict to the fact that spermine enhances LLPS, which means enhanced attraction.

6. When discussing the kinetics of LLPS, such as in Fig.S6, S4,S8, the authors plot the droplet diameter as a function of time to distinguish the effect of added SPM. However, this is not convincing. For example, in Fig.S6B, the plots show that with or without SPM, the tau protein LLPS shows similar trends, very close to each other. However, when looking at the images, one can clearly see that not only the size of the droplets, but also the number of them, i.e. the volume of the droplets is a better parameter to discuss this issue.

Reviewer #2

(Remarks to the Author)

Sun et al. investigated the role of spermine, a polyamine, in modulating protein condensation of Tau and α -synuclein (α S) and examined the associated functional implications for neurodegeneration. The authors first demonstrated that spermine can promote LLPS of both Tau and α -synuclein and enhancing protein mobility in the condensates using purified proteins. The authors then utilized diverse approaches, including TR-SAXS, NMR, and coarse-grained MD simulations, to investigate which regions of Tau and α S interact with spermine, as well as to elucidate how these interactions influence the structure and intermolecular interactions of each factor. Importantly, the authors demonstrated spermine-dependent reduction of aggregates in *C. elegans* and showed that this reduction relies on autophagic degradation. Finally, treatment with spermine in *C. elegans* models expressing Tau and α S significantly extended lifespan, improved motor function, and restored mitochondrial health, highlighting potential therapeutic applications for neurodegenerative diseases.

This study thoroughly investigates the specific effects of spermine on two aggregation-prone proteins, Tau and α S, both of which are crucial but distinct causative proteins in neurodegenerative diseases. Employing various physicochemical and computational methods, the authors clarify the detailed mechanisms by which spermine interacts with these proteins and regulates their LLPS. Additionally, using *C. elegans* as an in vivo model, they demonstrate spermine's effectiveness in preventing aggregate accumulation and even extending lifespan. These findings provide comprehensive insights into the beneficial effects of spermine and are thus highly significant and contribute to the development of this field. However, the experimental approaches used to evaluate autophagic activity appear somewhat primitive and thus require substantial improvement.

Major points

1) In the experiments shown in Figure 5, first verify whether RNAi is functioning effectively by performing Western blotting for each targeted factor.

2) Regarding the experiments presented in Figure 5C-D, use appropriate positive controls to confirm that RNAi targeting atg10, lc3, and lamp is effectively inhibiting autophagic activity. Next, statistically compare the effects of RNAi for atg10, lc3, and lamp with RNAi empty vector (ev) in the untreated control groups to demonstrate the role of autophagy in determining inclusion numbers. In the spermine (SPM)-treated groups, it remains unclear why RNAi of atg10 showed a significant difference, while RNAi of lc3 and lamp did not, given that Atg10 functions in a pathway similar to LC3. Clarify why this discrepancy exists.

3) Ideally, perform experiments using worms with ATG gene knockouts (KOs) rather than relying solely on RNAi. If obtaining knockout worms is challenging, conduct additional RNAi experiments targeting upstream autophagy components, such as FIP200, ATG13, and ATG14 (verification of RNAi efficiency is essential). This will clearly determine whether the reduction in inclusions mediated by SPM occurs through the autophagy pathway.

4) The observation that spermine increases FRAP recovery and reduces sensitivity to 1,6-HD suggests that spermine shifts the mechanism driving LLPS of Tau from hydrophobic to hydrophilic interactions. Discuss this possibility in detail.

Additionally, in the experiment shown in Figure 2D, was there any characteristic pattern in the types of amino acids within Tau that exhibited changes upon spermine titration?

Minor point

1) At lines 255 and 260, “Fig. 2I” should be “Fig. 2D”.

Reviewer #3

(Remarks to the Author)

Sun et al. set out on the non-trivial task of providing an atomistic to mesoscopic characterization of the role for spermine modulation of Tau and alpha-synuclein (a-syn) condensation in the test tube and a *C. elegans* model. The authors propose the experiments show that spermine extends lifespan and removes mobility impairment due to Tau and a-syn in the animal model. Furthermore, spermine promotes clearance of a-syn in the animal model through autophagy. Experiments on purified proteins from recombinant *E. coli* systems show that spermine promotes the liquid-like condensation of Tau and a-syn.

I am not an expert in *C. elegans* systems, so I hope another reviewer is an expert in this area. The sections pertaining to the *C. elegans* systems were clearly presented for a general audience and the conclusions seemed to follow from the results, to the best of my understanding.

In line 76, the authors hypothesize that liquid condensates, not fibrils, are more adaptable to dynamic processes like autophagosome formation and expansion. It is not clear if proving this hypothesis is the main goal of the manuscript, but the results in the manuscript do not provide this to be true unequivocally.

In general, it seems that the experiments are carefully performed. The comparison of spermine effects on Tau and a-syn condensation side-by-side is interesting. However, this becomes distracting when the paper focuses on solely Tau (NMR investigation) or a-syn (C. elegans FRAP and autophagy investigation). I recognize that the authors reference a-syn NMR measurements from the literature, but it would be useful to have a graphical comparison to go along with the Tau results. No explanation is given why Tau experiments are not presented along with the a-syn results in the C. elegans results in Figures 4 and 5. I recognize that the results in figure 6 pertain to both Tau and a-syn animal models, but this only increases the expectation that a similar analysis for Tau should accompany Figures 4 and 5.

I am not convinced that the experiments in the paper fully support the model in Figure 7. A major question is why the red Tau cartoons were shown to interact with the autophagy process but not the black a-syn, when the results for autophagy were presented for a-syn.

The results section pertaining to the TR-SAXS data is very difficult to follow and it is not clear how the authors derive a difference in folding for Tau and a-syn from the data presented in the paper. There is also discussion of a barrier, but what that refers to is not clear from the discussion or the figures. It could be that an expert in SAXS would easily interpret the results, however, the results are not clearly interpretable to a more general Nature Communications audience.

The authors need to address the in vitro conditions for the various assays. I am concerned that performing the NMR measurements in a phosphate buffered saline system might interfere with the charged interactions being investigated. It would be helpful if control experiments showing fluid droplets and spermine effects were presented in this buffer system.

Also pertaining to the NMR measurements, the authors conclude there are no specific interactions between spermine and Tau, but the intensity decreases shown in Figure S11C very clearly quench the NMR signal at some sites but not others. For the chemical shift changes, there do seem to be changes biased toward the N-terminal region of Tau. These results seem inconsistent with the interpretation that the interaction is largely non-specific. Furthermore, a comparison of quenched sites and charge patterning for Tau would be a helpful addition to the interpretation of the NMR results.

For the coarse grained simulations, the use of the epsilon parameter is not clearly justified and will seem arbitrary to someone outside a computational or biophysics field, like someone interested in animal models, for example.

Overall, the paper is interesting, but warrants a significant rewrite and additional experiments before the broad conclusions in Figure 7 follow from the experiments and analysis in the paper.

More specific points follow, below, some of which address the more general comments already presented.

- Line 111, the authors should quantify what "no impact" means
- Line 137, the data from the 1,6HD assay does not support the statement that hydrophobic interactions play a role in the phase separation of tau droplets. How do the authors know that these are not two fundamentally different kinds of droplets? Is there evidence that all interactions are always taking place in the literature? Or could droplets be of two different species, where either only charged interactions or hydrophobic interactions facilitate droplet formation, despite being present in the same protein?
- Line 159, Fig S8 has no panel C
- Lines 222–228, is not written so that a non-expert in TRSAXS can interpret the result. How is 0.2 s time constant determined? It looks like the minimum Rg value, but how is that related to a time constant? What does a global conformational rearrangement mean? Is there any chance of tau dimers under these conditions? What is the initial step? What barrier are the authors referring to?
- Lines 231–233, it is not at all clear what the authors are referring to here. What is the minimum dimension? What does it mean to reduce in size due to a decrease in average dimensions and how does that relate to complete or partial folding? Supplemental Fig. S10, what does beginning assembly refer to here?
- Line 234, This paragraph seems to be re-discussing some of the results from the Tau-spermine SAXS data, with an added discussion of a-syn. This section is in general confusing and the link between the experimental data and conclusions regarding the differences in folding/not folding between Tau and a-syn are not clear to the reader.
- Line 255, should this refer to Figure 2D, not 2I?
- Line 312–314, the authors need to explain the results more clearly for the coarse grained simulations. What does it mean for the simulations that the red and blue lines converge for Tau but not a-syn? What is being optimized? The radius of gyration? What is the expected value and how is that determined?
- Line 319–320, how does a value of epsilon=0.15 provide a clear variation of contacts?
- Line 333, the conclusion for Tau should be separated from a-syn, the sentence implies there is a difference in n-terminal and c-terminal interactions for Tau, which the authors claim is uniform earlier in the paragraph
- Line 356–357, a statistical test is required to support the claim that Tau concentration increases in a spermine dependent manner
- Lines 598–614, help with understanding the TRSAXS data, but the results are presented in such a way that the conclusion is not obvious from the text of the results section.

Reviewer #4

(Remarks to the Author)

This manuscript presents an interdisciplinary investigation of the effects of the polyamine spermine on the condensation behavior, molecular conformation, and clearance of two aggregation-prone proteins, Tau and α -synuclein (α S). By integrating TR-SAXS, NMR spectroscopy, coarse-grained simulations, and in vivo models, the authors convincingly demonstrate divergent molecular responses to spermine between Tau and α S, and propose a functional role for spermine in promoting autophagic degradation of condensates.

Overall, this is a thoughtfully executed and broadly relevant study. The combination of orthogonal techniques is a particular strength, and the biological implications of the findings are significant, particularly for neurodegeneration research. The paper is generally well written, with figures that support the narrative effectively.

However, I offer the following major comments and minor suggestions that I believe could improve clarity, reproducibility, and mechanistic insights.

Major Comments

1. Clarity on Spermine's Role in Aggregation versus LLPS

While the authors show that spermine enhances LLPS and fluidity of α S and Tau condensates, it remains unclear whether this ultimately inhibits or promotes amyloid fibril formation over time. Given the neurodegenerative disease context, additional experiments or discussion clarifying whether spermine-treated droplets mature into fibrils—or remain reversible—would be helpful. Time-course ThT assays or EM images could help clarify this trajectory.

2. Validation of CG Simulations across Parameter Space

The coarse-grained model required ϵ tuning specifically for Tau (K18 region) to reproduce spermine-induced LLPS. While this is justified by experimental trends, the manuscript would benefit from more systematic validation of this parameter adjustment, for instance by comparing predicted R_g values across spermine ratios with the TR-SAXS data in a more quantitative fashion.

3. FRAP Quantification and Statistical Analysis

The FRAP data are visually presented and qualitatively interpreted, but key parameters like mobile fraction or half-time recovery ($t_{1/2}$) are not reported. Including these values with statistical comparison (e.g., mean \pm SD or SEM across droplets/worms) would strengthen claims about enhanced fluidity and dynamics due to spermine.

4. Charge Patterning vs. Net Charge in Driving LLPS

The work nicely discusses net charge per residue (NCPR), but does not explicitly analyze charge patterning, which recent literature shows can have critical effects on LLPS. Since Tau and α S differ in their charge patterning, especially in IDRs, a sequence-level analysis (e.g., kappa or linear charge segregation) would be informative.

5. Specificity of Autophagy Involvement

The data convincingly show spermine-driven condensate clearance is autophagy-dependent. However, do the authors rule out the proteasome pathway or assess if spermine also promotes ubiquitination? This distinction is important and should be clarified, or at least discussed.

6. Definition of 'Molecular Glue'

The title and abstract use the term "molecular glue," but this typically implies induced binding between two components via non-covalent bridging. While spermine acts via electrostatics, it is not strictly gluing Tau and α S together. A clearer operational definition in the Introduction would help avoid confusion with synthetic glues used in targeted protein degradation.

7. Integration of Multiscale Data

The manuscript spans NMR, SAXS, CG simulations, and in vivo phenotypes. A summary figure or table (perhaps in the

supplement) that aligns the findings across these techniques (e.g., R_g , FRAP mobility, contact maps, worm lifespan) would help readers see the full multiscale integration and reinforce the consistency of conclusions.

Minor Suggestions

1. Terminology consistency

Use either “coacervation” or “LLPS” consistently, unless specifically distinguishing between them.

2. pH specification

The NMR spectra were acquired at pH 6.3, but most in vitro and in vivo assays are near pH 7.4. It would help to discuss how this slight acidification might influence electrostatics or spermine interaction, especially for Tau.

3. In vivo degradation kinetics

In the worm experiments, do the authors know how long spermine-treated condensates persist before clearance? Even approximate degradation kinetics could contextualize the biological relevance.

4. 1,6-Hexanediol control

The supplementary data show sensitivity to 1,6-HD, but it would help to mention this in the main text, as it supports the material state of the condensates.

Version 1:

Reviewer comments:

Reviewer #1

(Remarks to the Author)

The authors have carefully addressed my comments raised in previous round on the LLPS phase behavior and the TR-SAXS measurements and data analysis. The corrections and explanations are reasonable. The sections related to the LLPS phase behavior and TR-SAXS data analysis and interpretation are clear now.

Reviewer #2

(Remarks to the Author)

The authors have addressed all of my concerns.

Reviewer #3

(Remarks to the Author)

My concerns have been addressed. However, the other three reviewers have brought up very important points and the manuscript should be accepted only after those reviewers have agreed that their concerns have been fully addressed.

Reviewer #4

(Remarks to the Author)

I appreciate the authors' thoughtful and thorough revisions in response to my earlier comments. The clarifications on condensate dynamics versus fibril formation, the additional controls (such as mass photometry, ThT assays, qPCR validation), and the removal of overstated interpretations have strengthened the manuscript. The discussion is now more balanced and consistent with the data.

I have only two minor suggestions for further improvement:

Balance of Tau vs α -synuclein presentation: While I understand the technical challenges, it would help to explicitly acknowledge in the Discussion that new NMR data were only generated for Tau, and that α -synuclein insights rely partly on prior studies. A schematic comparison (even adapted from literature) showing the parallels and differences in spermine interactions with Tau and α -synuclein would strengthen the side-by-side narrative.

Hypothesis framing: The manuscript now frames the condensate versus fibril hypothesis more cautiously, but I encourage the authors to state explicitly that their data are consistent with, rather than conclusive proof of, the idea that liquid-like condensates are more accessible to autophagic clearance.

Please find below the response the reviewers with comments in black and our replies in blue. The corresponding changes in the text are highlighted in red.

REVIEWER COMMENTS

Reviewer #1 (Remarks to the Author):

In this interesting research project, the authors discussed the effects of spermine on the conformation, interaction and LLPS of Alzheimer's Tau and Parkinson's alpha-Synuclein. The research design is consistent including utilizing the advanced experimental methods and simulations, and the results provide new insights for biomolecular condensation and neurodegeneration. My comments focus on the phase behavior (LLPS) and the TR-SAXS measurements and results.

Response: We thank the reviewer for the overall assessment and comments that greatly helped us to improve this manuscript.

Specific comments are listed below.

1.1. Line 134-135, Fig.1F, the authors claim "In the presence of 1,6-hexanediol, Tau droplets without spermine were completely dissolved, while Tau droplets with spermine remained intact (Fig.1F). "The statement for Tau droplets is not fully correct. As one can see in Fig.1F, with increasing spermine concentration, the number of droplets is clearly reduced. When further increasing spermine concentration, Tau droplets may also dissolve.

R1.1: We thank the reviewer for pointing this out. We agree that our original statement was not fully accurate and appreciate the opportunity to clarify. Indeed, as shown in Fig. 1F, while 1,6-hexanediol completely dissolves Tau droplets in the absence of spermine, increasing spermine concentrations result in a partial resistance to dissolution rather than complete stability. We also observe that at higher spermine concentrations, the number of droplets decreases, suggesting that excessive spermine may shift the balance between electrostatic and hydrophobic interactions in the condensates.

Accordingly, we have revised the statement in the manuscript (**Page 4-5, Line 136-143**) to more accurately reflect the observed trend:

"In the presence of 10% 1,6-hexanediol, Tau droplets formed without spermine were completely dissolved, while those formed with spermine exhibited partial resistance to dissolution and the number of droplets was reduced (Fig. 1F). This suggests that although spermine promotes droplet formation likely through electrostatic interactions, hydrophobic interactions remain essential for the overall phase behavior. This is evidenced by the continued, though attenuated, sensitivity to 1,6-hexanediol with the presence of spermine, suggesting a likely shift from hydrophobic toward electrostatic interactions."

1.2. Fig.1, the effect of adding spermine on the LLPS of Tau and alpha-S shows some unconventional results. First of all, the phase diagram in Fig.1C and 1I clearly indicates that with increasing SPM the LLPS region is expanded, that is as the authors claimed "SPM enhance LLPS". However, the Fluorescence results in Fig.1B and H also clearly indicates that the protein concentration difference in the droplets (dense phase) and the dilute phase in fact are reduced with increasing SPM concentration. These observations seem contradictory to each other.

R1.2: We thank the reviewer for this insightful comment. We have replaced the images in Fig. 1B and H with the original and unprocessed images. All images were acquired using identical laser power and detector gain settings; however, they were taken from different wells which may account for the slight differences in background intensity observed. We have elaborated on this point in the revised manuscript (**Page 30, Line 814-816**).

“All images were acquired using identical laser power and detector gain settings; however, images from different wells which may account for the slight differences in background intensity observed.”

1.3. Line 213-214 **“The Tau concentration was increased from 0 to 100 uM...”** is this true? In the methods section (line777), it states that the protein concentration is fixed to 100 um, with and without SPM.

R1.3: We thank the reviewer for pointing out this inconsistency, and we apologize for the confusion. The statement in the original Line 213–214 was incorrect. As correctly described in the Methods section (**Line 838**), the Tau concentration was fixed at 100 μM in all experiments, both with and without spermine (SPM). We have corrected this error in the revised manuscript to ensure consistency and clarity (**Page 8, Line 226**).

1.4. For TR-SAXS measurements and results in Fig.2. Fig.S9,S10. One suggestion is that in Fig.S9 and S10, maybe better using double log-scale to emphasize the low q region as both Guinier analysis and the I0 information mainly from this region. Currently, the Intensity was plotted in log-scale but the label “Intensity” maybe a mistake.

R1.4: We have now replaced the old Fig. S9 as the new Fig. S11 with a double log-scale plot as the reviewer suggested. This adjustment emphasizes the low-q region, which is important for Guinier analysis and accurate determination of I_0 . In the updated figure, each scattering profile has been normalized by its respective I_0 value and vertically shifted for clarity. We have also corrected the y-axis label to accurately reflect the new scaling.

Figure S11: TR-SAXS measurement and EOM fitting of Tau and α S in the absence and presence of spermine. A-B, Experimental TR-SAXS data points and EOM fitting (black) plots of Tau in the absence (A) and presence of spermine (B) are displayed. Experimental curves from early time points are shown red, to late time points in black, with a color gradient to indicate time evolution for in-between time points. C-D, Experimental TR-SAXS data points and EOM fitting (black) plots of α S in the absence (C) and presence of spermine (D) are displayed.

1.4a. Line 222-223, “measurement of Tau with spermine showed that the compaction of Tau with a decay constant of ~ 0.2 s”. It is not clear to me what this decay constant means and how this value is determined. Furthermore, in Line 224-225, “the time constant of tau collapse was of the same order as the initial stage”. Again, not sure how the initial stage is defined.

R1.4a: We thank the reviewer for these insightful comments. We agree that the original wording was unclear and may have caused confusion. To clarify, our TR-SAXS measurements indicate that Tau undergoes a rapid conformational compaction within approximately 0.2 seconds after spermine is added. Rather than fitting this behavior to a precise exponential decay with a defined time constant, we now describe this as an ultrafast compaction phase that is immediately observable in the radius of gyration (R_g) profile.

Regarding the term “initial stage,” we were referring to the earliest detectable phase of conformational change upon spermine mixing, which is characterized by this abrupt decrease in

R_g. To avoid ambiguity, we have revised the relevant sentences in the manuscript (Page 8, Line 233-238) to read:

“However, upon addition of spermine, Tau underwent a rapid compaction within ~0.2 seconds, as observed by a sharp decrease in *R_g*, followed by a secondary expansion into a globular conformation with an *R_g* of 68.9 Å (Fig. 2B). This suggests that the subsequent conformational rearrangement may be governed by intra- and intermolecular interactions that limit further collapse.”

1.4b. Line 231-233 “the shapes of the two SAXS profiles overlaid well when normalized by *I*₀ and *R_g*,...”. First of all, in Fig.S10, the plots were not normalized. Secondly, if the statement here is correct, that the “reduction in size is from the decrease in average dimension, rather than **complete or partially folding**”, Isn’t this controversy to the observed increase of *R_g*?

R1.4b: We thank the reviewer for this valuable observation. Upon careful review, we agree that the SAXS profiles presented in Fig. S10 were not normalized by *I*₀ and *R_g*, and therefore the statement in the manuscript claiming good overlay of normalized profiles was inaccurate and not supported by the figure. Moreover, we recognize that our interpretation suggesting that the reduction in size arises solely from a decrease in average dimension—rather than from partial or complete folding—was potentially overstated and appears contradictory given the subsequent observed increase in *R_g*. To address these issues, we have removed the sentence in the manuscript to avoid overinterpretation and to maintain consistency with the experimental data. We believe this revision provides a clearer and more accurate presentation of the SAXS results.

1.4c. Increase of *R_g* for Tau over time, may indicate the aggregation instead of expansion. The current TR-SAXS measurements cannot exclude this option.

R1.4c: We thank the reviewer for highlighting this important consideration. To rule out the possibility that the observed increase in *R_g* arises from aggregation rather than conformational expansion, we conducted complementary mass photometry analysis, which showed that the Tau sample predominantly consists of monomeric species (~45 kDa) with no detectable aggregates (Fig. S12). Additionally, Thioflavin T (ThT) fluorescence assays (Fig. S10) indicate that spermine inhibits Tau aggregation, supporting the absence of aggregation within the sub-second timescale of our TR-SAXS measurements. Taken together, these results strongly suggest that the increase in *R_g* reflects a conformational rearrangement rather than early-stage aggregation.

Figure S12: Mass photometry measurement of Tau sample shows a predominant presence of monomeric Tau. Tau protein was prepared in 25 mM HEPES buffer (pH 7.4) and maintained on ice. A 2 μL of the sample was added to adjusted to a final concentration of 10 nM, sample was measured within 30 minutes of preparation.

Figure S10: Spermine suppresses amyloid fibrillation of Tau and α S in the condensate state. A, Thioflavin T (ThT) fluorescence assay monitoring Tau fibril formation over time in the absence and presence of SPM at molar ratios of 1:10 and 1:50 (Tau:SPM). (B-D) Quantification of kinetic parameters from (A), including lag time (B), half-time (C), and ThT plateau intensity (D). (E) ThT kinetics of α S aggregation with or without SPM at the indicated ratios. (F-H) Quantification of α S aggregation kinetics showing lag time (F), half-time (G), and plateau intensity (H). Error bars represent standard deviations from five replicates. Buffer: 25 mM HEPES, pH 7.4, 10% PEG8000, 20 μ M ThT.

1.4d. From Line 208 to line 251, the description in these two paragraphs has some sentences repeated, such as “Experimental TR-SAXS data, depicted in red,...”. More concise is needed.

R1.4d: We thank the reviewer for pointing this out. We have carefully revised the text between Lines 208 and 251 to remove redundancies and improve clarity. Repetitive phrases such as “Experimental TR-SAXS data, depicted in red, ...” have been streamlined or eliminated, and overlapping content has been consolidated to present a more concise and coherent description of the TR-SAXS results.

1.4e. The conclusions in this section require more support or discussion. “Tau underwent rapid initial compaction followed by an increase in its R_g , with the initial stage indicating a barrier-limited collapse driven by intramolecular interactions, while the subsequent expansion may be driven by more favorable intermolecular interactions.” In this statement, a barrier-limited collapse needs to be proved, and the intramolecular interactions need to be discussed. For the subsequent increase of R_g , on the one hand, as indicated above, increase of R_g may be due to protein aggregation.

The authors need to demonstrate this is not this case. It is also not clear why “more favorable intermolecular interactions” would drive the expansion instead of aggregation. One would expect an intramolecular repulsion to cause the expansion.

R1.4e: We appreciate the reviewer’s careful and thoughtful reading of our conclusions. Based on the feedback, we have revised the relevant section of the Results to improve clarity and avoid overinterpretation.

Regarding the Barrier-limited collapse: we acknowledge that the concept “barrier-limited collapse” is not directly evidenced by the current data. To avoid overstatement, we have removed this phrasing and instead describe the observed rapid decrease in R_g as an initial compaction phase. We now reference known behavior of IDPs, like hnRNPA1, which can undergo fast conformational compaction then expansion upon salt mediated LLPS detected by TR-SAXS (*Nat Commun* 12, 4513 (2021)).

As for the subsequent increase in R_g , we reiterate our earlier response that the observed expansion is unlikely due to aggregation, supported by complementary mass photometry and ThT assays (Fig. S12 and S10). Regarding the driving forces behind this expansion, rather than “more favorable intermolecular interactions,” we now propose that a balance of weak intermolecular attractions and intramolecular repulsions may lead to a reorganization into more expanded conformations. This interpretation is consistent with known IDP behavior where electrostatic repulsion and transient intermolecular contacts dynamically modulate conformational ensembles (*Molecular Cell*, 82(12):2201-2214., *Science*, 356(6339):753-756.). We have updated the manuscript (**Page 9, Line 246-248**) accordingly to reflect these points with appropriate references and clarifications.

“Tau underwent rapid initial compaction followed by an increase in its R_g (Fig. 2B), while the subsequent expansion may be driven by weak intermolecular attractions and intramolecular repulsions.”

1.5. Fig.3C, Line 330-333, not clear how to reach this conclusion “whereas for Tau, Spermine helps in overall expansion of the chain which explains the increases in R_g .” Further, the authors concluded that “Spermine can be considered as a surfactant to coat the negatively charged regions of the chain and reduces attractive interactions ...” if this is true, then it is conflict to the fact that spermine enhances LLPS, which means enhanced attraction.

R1.5: We thank the reviewer for this thoughtful comment. To clarify the rationale behind our conclusion regarding spermine-induced expansion and its relationship to LLPS, we have revised the manuscript to more explicitly describe the dual role of spermine in modulating Tau conformations and intermolecular interactions.

Specifically, we have modified the following paragraph in the Results section (**Page 14-15, Line 431-437**):

“For Tau, while similar spermine-induced expansion and LLPS are observed, spermine primarily functions to neutralize the negatively charged amino acids throughout the entire sequence rather than targeting specific regions. This neutralization reduces the intramolecular electrostatic attractions and leads to molecular expansion. The resulting expansion, combined with diminished electrostatic repulsion between molecules due to charge neutralization, facilitates Tau LLPS by enabling more frequent and permissive intermolecular encounters.”

This revised explanation clarifies that spermine reduces both intramolecular attractions (leading to expansion and increased R_g) and intermolecular repulsion (facilitating LLPS). Thus, the observed enhancement of LLPS is not contradictory to the chain expansion but rather a consequence of spermine's role in globally modulating electrostatics.

1.6. When discussing the kinetics of LLPS, such as in Fig.S6, S4,S8, the authors plot the droplet diameter as a function of time to distinguish the effect of added SPM. However, this is not convincing. For example, in Fig.S6B, the plots show that with or without SPM, the tau protein LLPS shows similar trends, very close to each other. However, when looking at the images, one can clearly see that not only the size of the droplets, but also the number of them, i.e. the volume of the droplets is a better parameter to discuss this issue.

R1.6: We thank the reviewer for this valuable suggestion. We agree that total volume of droplet provide important information when assessing LLPS kinetics in the presence of spermine. To address this, we have now reanalyzed our microscopy data and quantified volume fraction over time. This analysis has been included in Fig. S5, S6 and S9.

Figure S6: Spermine accelerates Tau LLPS in vitro. A, Representative time lapse images of phase separated droplets in the 20 μM of Tau in the absence and presence of 100 μM spermine in the presence of 10% PEG. Scale bar in the images is 20 μm . B, Quantification of the condensates volume fraction ($n=3$) corresponding to A.

Reviewer #2 (Remarks to the Author):

Sun et al. investigated the role of spermine, a polyamine, in modulating protein condensation of Tau and α -synuclein (α S) and examined the associated functional implications for neurodegeneration. The authors first demonstrated that spermine can promote LLPS of both Tau and α -synuclein and enhancing protein mobility in the condensates using purified proteins. The authors then utilized diverse approaches, including TR-SAXS, NMR, and coarse-grained MD simulations, to investigate which regions of Tau and α S interact with spermine, as well as to elucidate how these interactions influence the structure and intermolecular interactions of each factor. Importantly, the authors demonstrated spermine-dependent reduction of aggregates in *C. elegans* and showed that this reduction relies on autophagic degradation. Finally, treatment with spermine in *C. elegans* models expressing Tau and α S significantly extended lifespan, improved motor function, and restored mitochondrial health, highlighting potential therapeutic applications for neurodegenerative diseases.

This study thoroughly investigates the specific effects of spermine on two aggregation-prone proteins, Tau and α S, both of which are crucial but distinct causative proteins in neurodegenerative diseases. Employing various physicochemical and computational methods, the authors clarify the detailed mechanisms by which spermine interacts with these proteins and regulates their LLPS. Additionally, using *C. elegans* as an in vivo model, they demonstrate spermine's effectiveness in preventing aggregate accumulation and even extending lifespan. These findings provide comprehensive insights into the beneficial effects of spermine and are thus highly significant and contribute to the development of this field. **However, the experimental approaches used to evaluate autophagic activity appear somewhat primitive and thus require substantial improvement.**

Response: We thank the reviewer for the constructive insight. We have improved manuscript based on your comments.

Major points

2.1) In the experiments shown in Figure 5, first verify whether RNAi is functioning effectively by performing Western blotting for each targeted factor.

R2.1: We thank the reviewer for this important point. Due to limitations in antibody availability and sensitivity against the targeted proteins, we were unable to validate RNAi efficiency by Western blotting. Instead, we established and assessed knockdown efficiency using qPCR to measure mRNA levels. The results confirmed significant reduction in transcript levels for each targeted gene upon RNAi treatment. These qPCR validation data have been included in the revised manuscript.

Figure. S22, qPCR analysis confirms the efficiency of RNAi, demonstrating that gene expression levels in RNAi-treated groups were reduced to less than 10% of those observed in control groups.

The results are referenced in the text accordingly.

“We fed these worms with different RNAi bacteria (empty vector, sms RNAi and smd RNAi) to silence the spermine synthase (SMS) and S-adenosylmethionine decarboxylase (SMD) genes and verified by qPCR (Fig. S22)”, (Page 18, Line 497-500).

“We used RNAi to selectively silence interested target genes (lc3 and lamp) and verified by qPCR (Fig. S22)”, (Page 18, Line 527-528).

2.2) Regarding the experiments presented in Figure 5C-D, use appropriate positive controls to confirm that RNAi targeting atg10, lc3, and lamp is effectively inhibiting autophagic activity. Next, statistically compare the effects of RNAi for atg10, lc3, and lamp with RNAi empty vector (ev) in the untreated control groups to demonstrate the role of autophagy in determining inclusion numbers. In the spermine (SPM)-treated groups, it remains unclear why RNAi of **atg10** showed a significant difference, while RNAi of **lc3 and lamp** did not, given that Atg10 functions in a pathway similar to LC3. Clarify why this discrepancy exists.

R2.2: We thank the reviewer for this insightful comment. In response:

Validation of Autophagy Inhibition: We confirmed the knockdown efficiency of atg10, lc3, and lamp via qPCR (Fig. S22). To verify functional suppression of autophagy, we included beclin-1 RNAi and empty vector as a positive control. Beclin-1 knockdown significantly increased α S inclusion formation (Fig. S24 and Fig. 5), supporting the validity of our RNAi-based approach.

Statistical Comparison: We now include statistical comparisons between each autophagy RNAi treatment and the empty vector control in untreated groups. These analyses (**updated in Fig. 5**) demonstrate that RNAi-mediated autophagy inhibition significantly increases inclusion numbers, underscoring the role of autophagy in α S condensate clearance.

Figure 5: Autophagy is essential for spermine-mediated α S degradation in *C.elegans* model of PD. A, Representative images of NL5901 worms treated with 500 μ M of SPM (right) or vehicle control (left) and fed with empty vector (ev, top), sms RNAi (middle), smd RNAi (bottom). B, Quantification of condensate of α S-YFP in NL5901 *C.elegans* that treated with empty vector, sms, smd RNAi and SPM or vehicle control. C, Representative images of NL5901 worms treated with 500 μ M of SPM (right) or vehicle control (left) and fed with atg-10 RNAi (top), lc3 RNAi (middle) and lamp RNAi (bottom). D, Quantification of condensate of α S-YFP in NL5901 *C.elegans* that treated with empty vector, atg-10 RNAi, lc3 RNAi, lamp RNAi and SPM or vehicle control. The scale bar is: 20 μ m. The data represent the mean \pm SEM. A paired t test was used in C-D (ns: not significant, * $P < 0.05$, ** $P < 0.01$, *** $P < 0.001$).

Discrepancy in SPM-treated Conditions: Although Atg10, LC3, and LAMP are components of the autophagy pathway, their roles occur at different stages. Atg10 functions early in autophagosome biogenesis, facilitating phagophore expansion, while LC3 and LAMP act at later stages—autophagosome maturation and lysosomal fusion, respectively. We propose that spermine stabilizes α S condensates in a dynamic, mobile state that preferentially interacts with early autophagy machinery (e.g., Atg10-associated complexes). This may explain why atg10 RNAi significantly impairs condensate clearance under SPM treatment, while lc3 and lamp RNAi show subtler effects. We have clarified this mechanistic distinction in the revised Discussion.

“Interestingly, in the spermine-treated groups, RNAi knockdown of atg10 resulted in a significant decrease in α S inclusion numbers, whereas knockdown of beclin, lc3 and lamp did not show comparable effects. This discrepancy may reflect the stage-specific roles of these autophagy

components. *Atg10* functions at an early step in autophagosome biogenesis by facilitating phagophore membrane expansion, whereas LC3 and LAMP are involved in later steps—autophagosome maturation and lysosome fusion, respectively. We propose that spermine stabilizes α S condensates in a highly mobile and dynamic state, thereby promoting interactions with early autophagy machinery such as the *Atg10*-associated complex. In contrast, the downstream components (LC3 and LAMP) may have reduced accessibility or engagement with these dynamic condensates during the early clearance phase. This stage-selective interaction may explain the stronger effect observed upon *atg10* knockdown and underscores the importance of early autophagic events in spermine-facilitated clearance of α S condensates.” (Page 26, Line 697-710)

2.3) Ideally, perform experiments using worms with ATG gene knockouts (KOs) rather than relying solely on RNAi. If obtaining knockout worms is challenging, conduct additional RNAi experiments targeting upstream autophagy components, such as FIP200, ATG13, and ATG14 (verification of RNAi efficiency is essential). This will clearly determine whether the reduction in inclusions mediated by SPM occurs through the autophagy pathway.

R2.3: We thank the reviewer for this valuable suggestion. Although we were unable to obtain knockout worm strains, we expanded our analysis to include RNAi *beclin-1*, a key upstream regulator of autophagy initiation in *C. elegans*. *Beclin 1* functions in the class III PI3K complex, which is essential for phagophore nucleation. These new results have been incorporated into the revised manuscript.

“RNAi knockdown of *beclin-1* significantly impaired spermine-mediated reduction of inclusions, further supporting the conclusion that spermine-induced clearance is autophagy-dependent. Knockdown efficiency was confirmed via qPCR (Fig. S24),”

Figure S24: SPM treatment does not change α S-YFP condensates upon *beclin-1* RNAi knockdown. A, Representative images of NL5901 *C. elegans* expressing α S-YFP in the body wall muscle, treated with *beclin-1* RNAi under control or SPM-treated conditions. B, Quantification of total α S-YFP fluorescence intensity in control and SPM-treated animals. Data are represented as mean \pm SD; no significant difference was observed between groups (ns, unpaired t-test).

2.4) The observation that spermine increases FRAP recovery and reduces sensitivity to 1,6-HD suggests that spermine shifts the mechanism driving LLPS of Tau from hydrophobic to hydrophilic interactions. Discuss this possibility in detail. Additionally, in the experiment shown in Figure 2D,

was there any characteristic pattern in the types of amino acids within Tau that exhibited changes upon spermine titration?

R2.4: We thank the reviewer for this insightful comment. The increased FRAP recovery and reduced sensitivity to 1,6-hexanediol (1,6-HD) upon spermine treatment suggest a shift in Tau LLPS from hydrophobic to electrostatic or hydrophilic interactions. Since 1,6-HD disrupts weak hydrophobic interactions, the resistance of spermine-induced condensates indicates that LLPS is no longer primarily driven by hydrophobic forces.

Spermine, a polycation, likely binds to negatively charged residues—especially in the N/C-terminal tail—reducing intramolecular repulsion and promoting chain expansion and flexibility. This is reflected in increased R_g and enhanced FRAP dynamics. The expanded conformation allows for more effective electrostatic interactions between molecules, driving LLPS.

In Figure 2D, NMR titration revealed chemical shift perturbations mainly in acidic and polar regions (rich in Glu, Asp, Ser) of N-terminal tail, consistent with spermine binding and supporting the mechanism of charge-based LLPS modulation.

Minor point

2.5) At lines 255 and 260, “Fig. 2I” should be “Fig. 2D”.

R2.5: Thank you. This has been corrected.

Reviewer #3 (Remarks to the Author):

Sun et al. set out on the non-trivial task of providing an atomistic to mesoscopic characterization of the role for spermine modulation of Tau and alpha-synuclein (a-syn) condensation in the test tube and a *C. elegans* model. The authors propose the experiments show that spermine extends lifespan and removes mobility impairment due to Tau and a-syn in the animal model. Furthermore, spermine promotes clearance of a-syn in the animal model through autophagy. Experiments on purified proteins from recombinant *E. coli* systems show that spermine promotes the liquid-like condensation of Tau and a-syn.

I am not an expert in *C. elegans* systems, so I hope another reviewer is an expert in this area. The sections pertaining to the *C. elegans* systems were clearly presented for a general audience and the conclusions seemed to follow from the results to the best of my understanding.

Response: We sincerely thank the reviewer for their thoughtful and constructive feedback. We appreciate the comment regarding the clarity and accessibility of the *C. elegans* sections for a broader audience. As noted, another reviewer with expertise in *C. elegans* has provided a detailed evaluation of those experiments. We have carefully addressed all comments from both reviewers to further strengthen the manuscript and ensure clarity, rigor, and accuracy in the interpretation of our *in vivo* findings.

3.1) In line 76, the authors hypothesize that liquid condensates, not fibrils, are more adaptable to dynamic processes like autophagosome formation and expansion. It is not clear if proving this hypothesis is the main goal of the manuscript, but the results in the manuscript do not provide this to be true unequivocally.

R3.1: We thank the reviewer for this insightful comment. We agree that direct, real-time in vivo validation of this hypothesis would require further investigation. Nonetheless, we now provide additional supporting evidence consistent with this model. In vitro ThT assays (Fig. S10) show that spermine suppresses fibril formation of both Tau and α S in a crowding condition, stabilizing them in dynamic, non/less-amyloid condensate states. Our FRAP experiments confirmed that spermine mobilizes α S/Tau condensates after the incubation. In *C. elegans*, spermine treatment maintained α S in a more dynamic, liquid-like state and reduced inclusion load via the autophagy pathway (Fig. 4). Together, these findings support the idea that promoting condensate states over fibrils enhances susceptibility to autophagic clearance.

3.2) In general, it seems that the experiments are carefully performed. The comparison of spermine effects on Tau and a-syn condensation side-by-side is interesting. However, this becomes distracting when the paper focuses on solely Tau (NMR investigation) or a-syn (*C. elegans* FRAP and autophagy investigation). I recognize that the authors reference a-syn NMR measurements from the literature, but it would be useful to have a graphical comparison to go along with the Tau results. No explanation is given why Tau experiments are not presented along with the a-syn results in the *C. elegans* results in Figures 4 and 5. I recognize that the results in figure 6 pertain to both Tau and a-syn animal models, but this only increases the expectation that a similar analysis for Tau should accompany Figures 4 and 5.

R3.2: We thank the reviewer for this thoughtful comment. Our goal was to provide a side-by-side comparison of spermine's effects on Tau and α -synuclein (α S) across molecular and organismal contexts. While we agree that a parallel analysis in the *C. elegans* system for Tau would strengthen the narrative, a key limitation is the absence of an established and visually trackable Tau model in *C. elegans*. This currently precludes direct in vivo comparisons like those performed for α S in Figures 4 and 5. Nonetheless, we have included data in Figures S25 and S26 demonstrating that spermine improves lifespan, rescues movement deficits, and ameliorates mitochondrial dysfunction in the BR5270 and VH255 *C. elegans* models of Alzheimer's disease. These BR5270 and VH255 strains express the pro-aggregation F3 Tau fragment with Δ K280 and the full-length tau352(WT), respectively. These results complement our molecular findings and support the conserved modulatory role of spermine across both Tau and α S proteinopathies.

To enhance clarity and cohesion, we have now included a new summary table (Table S1) that integrates our findings on spermine's modulation of Tau and α S at both molecular (e.g., NMR and ThT) and organismal levels. We have also expanded the Discussion below to more explicitly align our Tau NMR data with published α S NMR results. We hope these additions help provide a clearer comparative framework for the reader.

Table S1. Summary of Multiscale Findings Across Experimental Techniques

Technique	Molecule	Experimental readout	Interpretation
NMR	Tau / α S	Residue-level chemical shift changes	SPM modulates electrostatics to favor Tau and α S LLPS
TR-SAXS	Tau / α S	R_g changes within 1s	SPM expands Tau and α S behavior
CG Simulation	Tau / α S	Contact map	Supports enhanced electrostatics intermolecular interaction upon SPM
LLPS Assay	Tau / α S	Phase diagram	SPM promotes Tau and α S LLPS
ThT Assay	Tau / α S	Amyloid fibril kinetics	SPM inhibits Tau and α S fibrillation
RNAi	α S	α S condensate	SPM facilitates α S condensates degradation via autophagy

FRAP	Tau / α S	Fluorescence recovery	SPM enhances Tau (in vitro) and α S (in vitro and vivo) mobility
Lifespan Assay	Tau / α S	Lifespan	SPM prolongs the lifespan of AD and PD C.elegans model
Behavior Assay	Tau / α S	Bend and head swing frequency	SPM improves fitness in AD and PD C.elegans model

3.3) I am not convinced that the experiments in the paper fully support the model in Figure 7. A major question is why the red Tau cartoons were shown to interact with the autophagy process but not the black a-syn, when the results for autophagy were presented for a-syn.

R3.3: We thank the reviewer for pointing out this important inconsistency. The original schematic in **Figure 7** was intended as a conceptual summary of our working model. However, we agree that the experimental evidence for autophagy involvement is currently limited to α S, based on our in vivo data in the *C. elegans* model. While both Tau and α S showed similar spermine-induced condensate dynamics in vitro, the link between Tau and autophagic clearance remains hypothetical at this stage.

In response, we have revised Figure 7 to more accurately reflect the data. Specifically, autophagy interactions are now illustrated only for α S, where direct evidence exists. The pathway involving Tau is now represented with dashed arrows to indicate that this part of the model is speculative and based on inference rather than direct experimental validation. We hope this change improves the clarity and integrity of the figure in relation to the presented data.

[Figure Redacted]

Figure 7: Schematic representation of the proposed mechanism for amyloid LLPS and degradation. Based on our observation, we propose a model for mobile amyloid condensate degradation via autophagosome expansion, which is initiated through spermine induced LLPS. Under physiological conditions, Tau and α S do not undergo phase separation but form less dynamic solid-like condensates. In contrast in the presence of spermine, Tau and α S both form dynamic liquid-like condensate with high mobility. These protein-dense condensates are

interconnected by weak intermolecular interactions, making them more readily recognized and fused by autophagosome and finally degraded via lysosome.

3.4) The results section pertaining to the TR-SAXS data is very difficult to follow and it is not clear how the authors derive a difference in folding for Tau and α -syn from the data presented in the paper. There is also discussion of a barrier, but what that refers to is not clear from the discussion or the figures. It could be that an expert in SAXS would easily interpret the results, however, the results are not clearly interpretable to a more general Nature Communications audience.

R3.4: We appreciate the reviewer's comment regarding the clarity of the TR-SAXS section. In response to this and a similar concern raised by Reviewer 1, we have substantially revised the relevant section in the Results to enhance accessibility for a broader audience. Specifically, we have clarified how the changes in radius of gyration over time reflect distinct compaction behaviors of Tau and α -synuclein during phase transition. This has been described in Result section.

“However, upon addition of spermine, Tau underwent a rapid compaction within ~0.2 seconds, as observed by a sharp decrease in R_g , followed by a secondary expansion into a globular conformation with an R_g of 68.9 Å (Fig. 2B). This suggests that the subsequent conformational rearrangement may be governed by intra- and intermolecular interactions that limit further collapse” (Page 8, Line 233-238)

3.5) The authors need to address the in vitro conditions for the various assays. I am concerned that performing the NMR measurements in a phosphate buffered saline system might interfere with the charged interactions being investigated. It would be helpful if control experiments showing fluid droplets and spermine effects were presented in this buffer system.

R3.5: We thank the reviewer for raising this important point regarding buffer conditions in our NMR experiments and other in vitro experiments. We have summarized and compared different experimental in Table S1. We also agree that buffer composition can influence electrostatic interactions and potentially modulate the phase behavior of intrinsically disordered proteins such as Tau.

In our study, NMR experiments were performed in phosphate buffer (20 mM $\text{Na}_2\text{HPO}_4/\text{NaH}_2\text{PO}_4$, pH 6.3,) to provide minimal ionic strength while maintaining compatibility with NMR sensitivity and line width. This low-salt condition was chosen specifically to preserve electrostatic interactions, including those between Tau and spermine, without introducing the high ionic strength typical of phosphate-buffered saline (PBS), which might mask such interactions.

To directly address the reviewer's concern, we have now conducted control experiments to confirm that Tau still forms dynamic liquid droplets under these conditions and that spermine promotes LLPS in this phosphate buffer system. These results are presented in new **Figure S13**, where DIC microscopy images show robust droplet formation of Tau alone and enhanced condensation upon spermine addition in the same phosphate buffer used for NMR.

Together, these controls confirm that the effects observed in the NMR experiments reflect genuine spermine-induced modulation of Tau LLPS and are not artifacts of buffer interference.

Figure S13: Spermine increases the propensity of Tau LLPS in phosphate buffer (20 mM $\text{Na}_2\text{HPO}_4/\text{NaH}_2\text{PO}_4$, pH 6.3). A, Representative images of phase separated 20 μM Tau droplets in the absence and presence of 100 μM spermine. B, Quantification of droplet volume fraction showing that spermine increases Tau LLPS in phosphate buffer used for NMR measurement. Buffer condition: 20 mM $\text{Na}_2\text{HPO}_4/\text{NaH}_2\text{PO}_4$, pH 6.3, 10% PEG8000.

3.6) Also pertaining to the NMR measurements, the authors conclude there are no specific interactions between spermine and Tau, but the intensity decreases shown in Figure S11C very clearly quench the NMR signal at some sites but not others. For the chemical shift changes, there do seem to be changes biased toward the N-terminal region of Tau. These results seem inconsistent with the interpretation that the interaction is largely non-specific. Furthermore, a comparison of quenched sites and charge patterning for Tau would be a helpful addition to the interpretation of the NMR results.

R3.6: We thank the reviewer for pointing out the possible interaction site(s) between Tau and spermine. We certainly agree with the reviewer that some of the NMR signals at certain sites in the protein sequence are indeed decreased in the presence of spermine. This effect is rather modest (on an average a 0-40% decrease from a 10-fold excess of spermine relative the protein concentration, but 20-60% in certain areas) and distributed at several regions all over the sequence, perhaps more pronounced in the N- and C-termini and in the negatively charged N-terminal part. It seems like there are some specificities, but rather weak interactions. In addition, we do not have information of all residues due to spectral overlap. With only one titration step, the signal decrease due to a concentration-dependence effect is challenging to follow, but, with the current data set we have now included a smoothing function using the median to be able to visualize any interaction pattern more clearly. A direct comparison between the Tau sequence, including the charge pattern, has also been added. Further, the interpretations and conclusions are accordingly updated in the text.

“However, despite a general intensity loss of approximately 20 to 60%, particularly more pronounced in the N- and C-terminal regions, no clear binding site(s) are observed. Instead, a diffusive effect is seen across several residues within Tau. Interestingly, from the chemical shift

changes a more pronounced effect seems to be in the negatively charged N-terminal region of Tau, suggesting the importance electrostatic interactions with spermine.” (Page 9, Line 269-274)

“Another observation includes less of a decrease of the crosspeak signal intensities for the shorter K18 fragment compared to the full-length Tau protein, which supports the suggestion of substantial electrostatic interactions (Fig. S14C, D). Taken together, this high-resolution structural information brings insights into how spermine interacts with Tau at the monomeric level. These interactions effects are likely diffusive, potentially due to a weak binding with dominantly electrostatic interactions.” (Page 10, Line 286-290)

3.7) For the coarse grained simulations, the use of the epsilon parameter is not clearly justified and will seem arbitrary to someone outside a computational or biophysics field, like someone interested in animal models, for example.

R3.7: We thank the reviewer for pointing this out. In our simulations, we employ the HPS (Hydrophobicity Scale) coarse-grained force field, as introduced in a previous literature (*PLoS Comput Biol* **2018**, *14* (1), e1005941). The epsilon (ϵ) parameter in the model is a free parameter and sets the strength of the short-range pairwise interactions between coarse-grained residues. This value was optimized for the HPS model to reproduce experimentally determined radius of gyration (R_g) values of a set of intrinsically disordered proteins (IDPs). So, for a specific IDP, the value is not universal and can typically be calibrated based on existing experimental data.

In our recent work of investigating phase separation of α S (*Adv. Sci.* **2024**, *11*, 2308279), we further refined the model by incorporating backbone angle and dihedral potentials, which required re-optimizing the ϵ values. Here by utilizing TR-SAXS experiment, we found an ϵ of approximately 0.09 kcal/mol best match the R_g of α S and Tau at the experimental ionic strength without spermine.

However, we acknowledge that in vivo conditions differ from in vitro settings, especially due to molecular crowding, which can effectively strengthen intermolecular interactions. This makes it difficult to choose a single ϵ that fits all biological contexts. To address this, we do not rely on a single ϵ value; instead, we systematically explore a range of ϵ values to ensure that our conclusions are not specific to one parameter setting. For example, Fig. 3A shows how R_g changes with and without spermine across several ϵ values, and we confirmed that the qualitative trends (e.g. spermine-induced expansion) we reported are consistent across this range.

We have clarified this rationale in the revised manuscript to improve accessibility for readers unfamiliar with coarse-grained modeling as

“As shown in Fig. 3A, we scanned a variety of hydrophobic interaction strengths, denoted by ϵ , which controls the strength of short-range pairwise interactions between residues. While ϵ is a model parameter, it is typically calibrated to reproduce experimental observables such as R_g , and trends are confirmed across a range of values to ensure robustness. Using TR-SAXS experiment as guidance, we found an ϵ of approximately 0.09 kcal/mol best match the R_g of α S and Tau at the experimental ionic strength without spermine.” (Page 11-12, Line 319-325)

We also note that we found one single ϵ value for Tau cannot reproduce the spermine-induced phase separation. Motivated by K18’s aggregation-prone nature, we specifically tune the ϵ value of the hydrophobic interactions within the K18 region to capture the experimental trend. Please refer to the reply R4.2 to Reviewer 4’s question for further details.

3.8) Overall, the paper is interesting, but warrants a significant rewrite and additional experiments before the broad conclusions in Figure 7 follow from the experiments and analysis in the paper.

R3.8: We thank the reviewer for their overall assessment and constructive suggestions. In response to this and the specific points raised, we have undertaken a major revision of the manuscript to strengthen both the clarity and the experimental support for our conclusions. These changes include:

- (1) Substantial revision of the TR-SAXS Results section and figures, improving clarity and accessibility for a broader readership, as also recommended by other reviewers.
- (2) Addition of new experimental data, including:
 - a. Mass photometry and ThT kinetic assays (Fig. S12 and S10), demonstrating that spermine inhibits fibrillation of both Tau and α S.
 - b. qPCR validation of RNAi efficiency (Fig. S22).
 - c. Beclin-1 RNAi experiments (Fig. S24) to further support the role of autophagy in clearance.
 - d. Quantification of the mobile fraction in FRAP experiments for α S (Figs. S7 and S21).
- (3) Revision of the schematic model (Fig. 7) to more accurately reflect the current experimental evidence and clarify the mechanistic distinctions.

We are grateful for the reviewer's feedback, which has significantly improved the rigor, coherence, and presentation of our manuscript.

More specific points follow, below, some of which address the more general comments already presented.

3.9) Line 111, the authors should quantify what "no impact" means

R3.9: We thank the reviewer for this helpful suggestion. "no impact" refers that spermine did not change the K18 LLPS phase diagram in the presence of different percent of PEG.

In the revised manuscript, we have rewritten that "In contrast, spermine did not alter the LLPS behavior of K18 in the presence of either 10% PEG (Fig. S3A–B) or 7.5% PEG (Fig. S3C–D)."

3.10) Line 137, the data from the 1,6HD assay does not support the statement that hydrophobic interactions play a role in the phase separation of tau droplets. How do the authors know that these are not two fundamentally different kinds of droplets? Is there evidence that all interactions are always taking place in the literature? Or could droplets be of two different species, where either only charged interactions or hydrophobic interactions facilitate droplet formation, despite being present in the same protein?

R3.10: We appreciate the reviewer's insightful comment regarding the interpretation of the 1,6-hexanediol (1,6-HD) assay and the potential for multiple types of Tau condensates with distinct interaction profiles. This is an important point, and we have expanded our discussion to clarify the interpretation and limitations of our observations.

As we also noted in our reply to Reviewer 1 (R1.1), Tau droplets formed with spermine exhibit partial resistance to 1,6-HD-induced dissolution, rather than full insensitivity. This suggests that while electrostatic interactions, enhanced by charge neutralization from spermine, play a major role in driving LLPS, hydrophobic interactions likely still contribute to the overall droplet stability. The attenuated but not eliminated sensitivity to 1,6-HD supports the idea that hydrophobic contacts remain relevant, even in spermine-stabilized droplets.

We agree with the reviewer that the possibility of distinct droplet species with different interaction mechanisms cannot be ruled out. However, our current data support a continuum model in which both electrostatic and hydrophobic interactions are present and their relative contributions vary depending on the conditions (e.g., spermine concentration). This view is also consistent with prior studies showing in general IDP phase separation might be modulated by ionic strength, RNA, and crowding agents, all of which shift the balance between different interaction types rather than inducing entirely distinct droplet species.

We have modified the main text as

“In the presence of 10% 1,6-hexanediol, Tau droplets formed without spermine were completely dissolved, while those formed with spermine exhibited partial resistance to dissolution and the number of droplets was reduced. This suggests that although spermine promotes droplet formation likely through electrostatic interactions, hydrophobic interactions remain essential for the overall phase behavior. This is evidenced by the continued, though attenuated, sensitivity to 1,6-hexanediol with the presence of spermine, suggesting a likely shift from hydrophobic toward electrostatic interactions.” (Page 4-5, Line 136-143)

3.11) Line 159, Fig S8 has no panel C

R3.11: Thank you. This has been corrected.

3.12) Lines 222–228, is not written so that a non-expert in TRSAXS can interpret the result. How is 0.2 s time constant determined? It looks like the minimum R_g value, but how is that related to a time constant? What does a global conformational rearrangement mean? Is there any chance of tau dimers under these conditions? What is the initial step? What barrier are the authors referring to?

R3.12: We thank the reviewer for these insightful comments. We agree that the original wording was unclear and may have caused confusion. To clarify, our TR-SAXS measurements indicate that Tau undergoes a rapid conformational compaction within approximately 0.2 seconds after spermine is added. Rather than fitting this behavior to a precise exponential decay with a defined time constant, we now describe this as an ultrafast compaction phase that is immediately observable in the radius of gyration (R_g) profile.

Regarding the term “initial stage,” we were referring to the earliest detectable phase of conformational change upon spermine mixing, which is characterized by this abrupt decrease in R_g . To avoid ambiguity, we have revised the relevant sentences in the manuscript to read:

“Upon addition of spermine, Tau underwent a rapid compaction within ~0.2 seconds, as observed by a sharp decrease in R_g , followed by a secondary expansion into a globular conformation with an R_g of 68.9 Å (Fig. 2B). This suggests that the subsequent conformational rearrangement may

be governed by intra- and intermolecular interactions that limit further collapse.” (Page 8, Line 233-238)

Regarding the reviewer's question about dimer formation, we conducted mass photometry measurements of Tau, which confirmed that Tau remains monomeric (~45 kDa) during the TR-SAXS measurement timeframe (now shown in Fig. S12) and spermine inhibited the Tau fibrillation (ThT kinetics, now shown in Fig. S10).

Regarding the “energy barrier”, we have removed this term.

3.13) Lines 231–233, it is not at all clear what the authors are referring to here. What is the minimum dimension? What does it mean to reduce in size due to a decrease in average dimensions and how does that relate to complete or partial folding? Supplemental Fig. S10, what does beginning assembly refer to here?

R3.13: We thank the reviewer for raising this point, which was also highlighted by Reviewer 1. We now removed these sentences to avoid over interpretation.

3.14) Line 234, This paragraph seems to be re-discussing some of the results from the Tau-spermine SAXS data, with an added discussion of a-syn. This section is in general confusing and the link between the experimental data and conclusions regarding the differences in folding/not folding between Tau and a-syn are not clear to the reader.

R3.14: We thank the reviewer for this helpful comment. To improve clarity and avoid redundancy, we have revised this paragraph substantially. We have removed repeated descriptions of the Tau–spermine SAXS results and instead provided a concise summary with a clearer comparison to α S.

3.15) Line 255, should this refer to Figure 2D, not 2I?

R3.15: Thank you. This has been corrected.

3.16) Line 312–314, the authors need to explain the results more clearly for the coarse-grained simulations. What does it mean for the simulations that the red and blue lines converge for Tau but not a-syn? What is being optimized? The radius of gyration? What is the expected value and how is that determined?

R3.16: The red and blue lines in Fig. 3A represent the radii of gyration of α S and Tau in the presence and absence of spermine, respectively, across a range of ϵ values (which modulate hydrophobic interaction strength). These simulations are not for the purpose of optimizing ϵ , but rather exploring how spermine-induced conformational changes behave across a spectrum of interaction strengths. This allows us to assess the robustness of observed trends under different crowding-like conditions. Please see also reply R3.7 for our motivation of scanning a variety of ϵ values due to the varying crowding conditions in vitro and in vivo.

For α S, R_g increases consistently with spermine regardless of ϵ , indicating strong electrostatic interactions with spermine across conditions. For Tau, however, at higher ϵ values, the R_g values in the presence and absence of spermine become more similar. So, the red and blue lines appear to converge, not because there is a true convergence in behavior, but because spermine has less impact when Tau is already in a more collapsed state. This reduced sensitivity likely results from decreased accessibility of interaction sites in compact conformations.

We have clarified this explanation in the revised main text to better reflect the distinction and remove any ambiguity about what is being optimized or interpreted.

“Consistent with TR-SAXS measurements (Fig. 2B-C), our CG simulations demonstrate spermine-induced expansion of both α S and Tau, as reflected by the increasing R_g values in presence of spermine. Interestingly, at higher ϵ values—intended to mimic increased cellular crowding—this expansion effect persists for α S but is diminished for Tau. This difference likely arises from the higher net charge density of α S, which promotes stronger electrostatic binding with spermine. In contrast, Tau, which adopts a more collapsed conformation under stronger crowding conditions, becomes less responsive to spermine, likely due to reduced accessibility of interaction sites.” (Page 12, Line 335-342)

3.17) *- Line 319–320, how does a value of $\epsilon=0.15$ provide a clear variation of contacts?

R3.17: The choice of $\epsilon=0.15$ kcal/mol reflects the crowding conditions used in our LLPS simulations and is therefore close to the conditions under which phase separation is observed experimentally. We selected this value to present the contact analysis in Fig. 3B because it balances two needs: first it corresponds to a biologically relevant crowded environment where LLPS occurs, and second it provides sufficient intermolecular interaction strength to clearly visualize spermine-residue contacts in single-chain simulations. At lower ϵ values, the protein conformations are more expanded and dynamic, leading to less interpretable contact profiles with larger errors. At $\epsilon=0.15$, sufficient number of contacts were observed, allowing clearer identification of interaction trends.

We have clarified this rationale in the revised text.

“Although lower ϵ values better reproduce the experimentally observed R_g of Tau and α S, we show the results in Fig. 3B using simulations at $\epsilon=0.15$ kcal/mol, as such a stronger interaction strength mimics the crowding condition used in our LLPS simulations and allows for clearer visualization of spermine-residue contacts under more compact conformations.” (Page 12, Line 354-358)

3.18) Line 333, the conclusion for Tau should be separated from a-syn, the sentence implies there is a difference in n-terminal and c-terminal interactions for Tau, which the authors claim is uniform earlier in the paragraph

R3.18: We thank the reviewer for pointing out this inconsistency in discussion. We have removed the sentence implying regional differences in Tau interactions and revised the paragraph to clearly state that spermine interacts more uniformly across the Tau chain, consistent with earlier statements and experimental data. The conclusion for Tau has also been separated from that of α S to avoid conflation.

3.19) Line 356–357, a statistical test is required to support the claim that Tau concentration increases in a spermine dependent manner

R3.19: We thank the reviewer for this suggestion. To statistically assess the claim that Tau density increases with spermine concentration, we performed a bootstrap analysis. Using the experimental density values and their associated errors, we generated 10,000 synthetic datasets by sampling from Gaussian distributions centered at the measured densities. Each dataset was fitted with a linear regression, and we compiled the distribution of slopes as shown in the attached

figure. We found that more than 95% of the bootstrapped slopes were positive, indicating a strong trend of increasing density with increasing spermine concentration. The simulations in one additional pH were requested by Reviewer 4. Furthermore, the one-sided 95% confidence lower bound for the slope were 0.07 and 0.4 for the two pHs, providing statistical support for a spermine-dependent increase in Tau density.

Figure S19: Tau phase separation at different pHs from coarse-grained simulations. (A) The densities of the Tau condensates at different pHs. (B) the histogram of linear fitting slope to the density-spermine concentration curve using a bootstrap test.

We have added the following discussion into the main text

“To statistically evaluate the trend, we performed a bootstrap analysis using the experimental density values and their errors. The results show that 95% of fitted slopes were positive, with a one-sided 95% confidence lower bound of 0.07 (see Fig. S19), supporting a spermine-dependent increase in Tau density.” (Page 13, Line 390-393)

3.20) Lines 598–614, help with understanding the TRSAXS data, but the results are presented in such a way that the conclusion is not obvious from the text of the results section.

R3.20: We thank the reviewer for this comment. We have substantially revised the TR-SAXS results section and reorganized the Figure S11.

Reviewer #4 (Remarks to the Author):

This manuscript presents an interdisciplinary investigation of the effects of the polyamine spermine on the condensation behavior, molecular conformation, and clearance of two aggregation-prone proteins, Tau and α -synuclein (α S). By integrating TR-SAXS, NMR spectroscopy, coarse-grained simulations, and in vivo models, the authors convincingly demonstrate divergent molecular responses to spermine between Tau and α S and propose a functional role for spermine in promoting autophagic degradation of condensates.

Overall, this is a thoughtfully executed and broadly relevant study. The combination of orthogonal techniques is a particular strength, and the biological implications of the findings are significant, particularly for neurodegeneration research. The paper is generally well written, with figures that support the narrative effectively.

However, I offer the following major comments and minor suggestions that I believe could improve clarity, reproducibility, and mechanistic insights.

Response: We thank the reviewer for their generous appreciation of our work.

Major Comments

4.1. Clarity on Spermine's Role in Aggregation versus LLPS

While the authors show that spermine enhances LLPS and fluidity of α S and Tau condensates, it remains unclear whether this ultimately inhibits or promotes amyloid fibril formation over time. Given the neurodegenerative disease context, additional experiments or discussion clarifying whether spermine-treated droplets mature into fibrils—or remain reversible—would be helpful. Time-course ThT assays or EM images could help clarify this trajectory.

R4.1: We thank the reviewer for raising this important point. To address this, we performed ThT fluorescence aggregation assays to assess the impact of spermine on Tau and α S fibrillization within condensates. As shown in Fig. S10, spermine progressively suppressed ThT fluorescence intensity and delayed the lag time over time for Tau (Fig. S10A-D) and totally prevented ThT intensity for α S (Fig. S10E-H). We have added these results and interpretation to the revised Results sections.

“To further assess the impact of spermine on Tau and α S fibrillization within condensates, we performed ThT fluorescence kinetics assay. As shown in Fig. S10, spermine reduced both Tau and α S fibrillization in a dose-dependent manner. For Tau, spermine increased the lag time and half-time of aggregation while decreasing the final ThT intensity, indicating delayed and reduced fibril formation (Fig. S10A–D). Similarly, spermine profoundly inhibited α S aggregation, as shown by the markedly extended lag time and reduced ThT plateau values at higher spermine concentrations (Fig. S10E–H). These results demonstrate that spermine acts as an inhibitor of amyloid fibril formation for both Tau and α S.” (Page 5, Line 171-179)

Figure S10. Spermine suppresses amyloid fibrillation of Tau and α S in the condensate state. A, Thioflavin T (ThT) fluorescence assay monitoring Tau fibril formation over time in the absence and presence of SPM at molar ratios of 1:10 and 1:50 (Tau:SPM). (B-D) Quantification of kinetic parameters from (A), including lag time (B), half-time (C), and ThT plateau intensity (D). (E) ThT kinetics of α S aggregation with or without SPM at the indicated ratios. (F-H) Quantification of α S aggregation kinetics showing lag time (F), half-time (G), and plateau intensity (H). Error bars represent standard deviations from five replicates. Buffer: 25 mM HEPES, pH 7.4, 10% PEG8000, 20 μ M ThT.

4.2. Validation of CG Simulations across Parameter Space

The coarse-grained model required ϵ tuning specifically for Tau (K18 region) to reproduce spermine-induced LLPS. While this is justified by experimental trends, the manuscript would benefit from more systematic validation of this parameter adjustment, for instance by comparing predicted R_g values across spermine ratios with the TR-SAXS data in a more quantitative fashion.

R4.2: We thank the reviewer for the thoughtful suggestion. As discussed in the manuscript, ϵ -tuning was necessary for the K18 region of Tau, as the use of a uniform ϵ for all residues in the coarse-grained model failed to reproduce the spermine-induced condensation behavior specific to Tau. Given that the K18 region is known to be involved in aggregation whereas the coarse-grained model may inadequately represent the cooperativity required for hydrophobic cluster formation, we increased the ϵ for hydrophobic residues within the K18 region to better account for these interactions.

We had systematically evaluated the impact of this adjustment on Tau's overall behavior, which was not shown previously. Specifically, we tested ϵ values of 0.2, 0.3, and 0.5 kcal/mol for the hydrophobic residues within K18 region and, for each of these, varied the ϵ values of the remaining residues in Tau. For all systems, we computed the R_g for both the full-length Tau protein and the K18 region individually, as shown in the new Fig. S17 (attached below). We also compared these results to simulations in which a uniform ϵ value was used for all residues. These data demonstrate that increasing ϵ for specific residues within the K18 region reduces the R_g of that region, while the R_g of full-length Tau does not show significant change by these local adjustments.

The revised manuscript now includes the following clarification:

"For Tau, the current CG model does not reproduce spermine-dependent phase separation as observed in the experiment. The co-existence (slab) simulations showed no spermine dependence in condensate formation (Fig. S16). Motivated by K18's aggregation-prone nature, we increased ϵ only for interactions between hydrophobic amino acids in the K18 region (residues 244-372). Simulations across various interaction strengths show that K18 compaction increases, while the full-length Tau's R_g remains largely unaffected (Fig. S17). Based on these results, all simulations of Tau have been performed with an increasing ϵ of 0.5 kcal/mol for the hydrophobic residues within K18 region." (Page 12, Line 326-334)

Figure S17: Variation of R_g when changing ϵ values in Tau simulations. R_g of full-length Tau are shown in (A) and the K18 segment in (B). The black dots represent results from simulations with a uniform ϵ whereas the other dots represent simulations where the ϵ for interactions between the hydrophobic residues within the K18 segment has been set to a larger value shown in the figure legend.

4.3. FRAP Quantification and Statistical Analysis

The FRAP data are visually presented and qualitatively interpreted, but key parameters like mobile fraction or half-time recovery ($t_{1/2}$) are not reported. Including these values with statistical comparison (e.g., mean \pm SD or SEM across droplets/worms) would strengthen claims about enhanced fluidity and dynamics due to spermine.

Answer: We thank the reviewer for this valuable suggestion. In response, we have quantified the FRAP recovery curves by calculating the mobile fraction for each condition. These values have now been reported in the revised manuscript as mean \pm SD. The results confirm that spermine significantly enhances Tau droplet fluidity, consistent with our qualitative observations. The relevant data and statistical analysis have been added to revised Figure S7 and Figure S21.

Figure S7: Quantification of the mobile fraction from FRAP experiments for Tau with or without SPM(A) corresponding to Figure 1D and α S with or without SPM (B) corresponding to Figure 1J.

Figure S21. Quantification of the mobile fraction from FRAP experiments for SPM treated *C.elegans* at day7 (A) corresponding to Figure 4D and day15 (B) corresponding to Figure 4F.

4.4.Charge Patterning vs. Net Charge in Driving LLPS

The work nicely discusses net charge per residue (NCPR), but does not explicitly analyze charge patterning, which recent literature shows can have critical effects on LLPS. Since Tau and α S differ in their charge patterning, especially in IDRs, a sequence-level analysis (e.g., kappa or linear charge segregation) would be informative.

R4.4: We thank the reviewer for this helpful suggestion. We agree that even with similar net charge per residue (NCPR), differences in charge patterning can influence LLPS. Sequences with more blocky distributions of charge may promote stronger interactions and enhance phase separation. In the absence of spermine, the κ values for α S and Tau are 0.172 and 0.184, respectively. Both are relatively low, indicating well-mixed charge distributions.

To explore the potential influence of spermine on charge patterning, we performed a simulation-based test in which we gradually substituted negatively charged residues (D/E) with neutral residues (N) to mimic spermine neutralization. For each level of substitution, we generated 100 random variants by selecting different substitution sites, and computed the mean and standard deviation of the resulting κ values. As shown in the new Fig. S18 (attached), the κ values showed only limited variation across all substitution levels. These results suggest that, although charge patterning is an important determinant of LLPS in general, it may not be a dominant factor in modulating Tau and α S behavior in our spermine titration experiments.

We have added the following text to the revised manuscript to address this point:

“Beyond net charge, global charge patterning may also influence conformational responses to spermine. We quantified charge patterning using the κ parameter, which measures how mixed or segregated charged residues are along the sequence. Both α S and Tau exhibit low κ values (~0.17 to 0.18), indicating well-mixed charge distributions. To mimic spermine-induced neutralization, we performed in silico substitution of acidic residues with neutral ones and found that κ values remain largely unchanged across substitution levels (Fig. S18). These results suggest that charge patterning is unlikely to be the primary factor driving the distinct global conformational responses of α S and Tau to spermine.” (Page 12, Line 342-351)

Figure S18: Effect of in silico neutralization of acidic residues on charge patterning (κ) for α S (A) and Tau (B). Negatively charged residues (D and E) were progressively substituted with neutral residues (N) at random positions to mimic spermine-induced charge neutralization. For each level of substitution, 100 randomized sequences were generated and κ values were computed. The mean and standard deviation of κ are shown for each substitution level.

4.5. Specificity of Autophagy Involvement

The data convincingly show spermine-driven condensate clearance is autophagy-dependent. However, do the authors rule out the proteasome pathway or assess if spermine also promotes **ubiquitination**? This distinction is important and should be clarified, or at least discussed.

R4.5: We thank the reviewer for this important point. In our current study, we focused on evaluating the role of autophagy in spermine-driven α S condensate clearance and did not directly assess proteasome involvement or changes in ubiquitination. While our data support an autophagy-dependent mechanism, we agree that we cannot fully exclude the potential contribution of the proteasome pathway. We have added a statement to the Discussion

acknowledging this limitation and highlighting the need for future studies to determine whether spermine also modulates proteasome activity or α S ubiquitination.

“While our findings indicate that spermine-driven α S condensate clearance is primarily autophagy-dependent, we cannot exclude potential contributions from the ubiquitin–proteasome system. Further studies will be needed to determine whether spermine affects proteasome activity or α S ubiquitination, which could act in parallel or in coordination with autophagy to regulate α S turnover”. (Page 25, Line 690-695)

4.6. Definition of ‘Molecular Glue’

The title and abstract use the term “molecular glue,” but this typically implies induced binding between two components via non-covalent bridging. While spermine acts via electrostatics, it is not strictly gluing Tau and α S together. A clearer operational definition in the Introduction would help avoid confusion with synthetic glues used in targeted protein degradation.

R4.6: We thank the reviewer for this important clarification. We agree that the term “molecular glue” may be misleading, as it often implies non-covalent bridging between two proteins, as seen in synthetic molecules designed for targeted protein degradation.

To address this, we have revised the manuscript to avoid potential confusion and have included a clearer definition of our operational use of the term in the Introduction. Specifically, we now describe spermine as a modulator of heterotypic phase separation via electrostatic effects, rather than as a classical molecular glue. This has been elaborated in our manuscript”.

“While the term molecular glue is often used to describe small molecules that induce protein–protein interactions by directly bridging two components, here we use the term more broadly to describe the ability of spermine to modulate heterotypic interactions and promote co-condensation via charge neutralization. This usage is operational and refers to the enhancement of intermolecular association rather than a defined ternary complex formation.” (Page 3, Line 84-89)

4.7. Integration of Multiscale Data

The manuscript spans NMR, SAXS, CG simulations, and in vivo phenotypes. A summary figure or table (perhaps in the supplement) that aligns the findings across these techniques (e.g., R_g , FRAP mobility, contact maps, worm lifespan) would help readers see the full multiscale integration and reinforce the consistency of conclusions.

R4.7: We appreciate the reviewer’s suggestion to improve clarity and cohesion across the multiscale datasets presented in the manuscript. To address this, we have now included a new summary table (Table S1). This table highlights how each technique contributes to a unified interpretation of the molecular behavior and biological consequences of Tau and α S condensates modulated by spermine.

Table S1. Summary of Multiscale Findings Across Experimental Techniques

Technique	Molecule	Experimental readout	Interpretation
NMR	Tau / α S	Residue-level chemical shift changes	SPM modulates electrostatics to favor Tau and α S LLPS
TR-SAXS	Tau / α S	R_g changes within 1s	SPM expands Tau and α S behavior

CG Simulation	Tau / α S	Contact map	Supports enhanced electrostatics intermolecular interaction upon SPM
LLPS Assay	Tau / α S	Phase diagram	SPM promotes Tau and α S LLPS
ThT Assay	Tau / α S	Amyloid fibril kinetics	SPM inhibits Tau and α S fibrillation
RNAi	α S	α S condensate	SPM facilitates α S condensates degradation via autophagy
FRAP	Tau / α S	Fluorescence recovery	SPM enhances Tau (in vitro) and α S (in vitro and vivo) mobility
Lifespan Assay	Tau / α S	Lifespan	SPM prolongs the lifespan of AD and PD C.elegans model
Behavior Assay	Tau / α S	Bend and head swing frequency	SPM improves fitness in AD and PD C.elegans model

Minor Suggestions

4.8. Terminology consistency

Use either “coacervation” or “LLPS” consistently, unless specifically distinguishing between them.

Answer: We thank the reviewer for this helpful suggestion. We have revised the manuscript to use the term “LLPS” consistently throughout.

4.9. pH specification

The NMR spectra were acquired at pH 6.3, but most in vitro and in vivo assays are near pH 7.4. It would help to discuss how this slight acidification might influence electrostatics or spermine interaction, especially for Tau.

R4.9: As the NMR spectra were acquired at pH 6.3, our original simulation used a histidine charge of +0.5 to remain consistent with this condition. This approach accounts for the partial protonation of histidine side chains at pH 6.3, based on their pKa values. To specifically address the concern regarding the difference in pH between the NMR conditions (pH 6.3) and physiological or in vitro assay conditions (pH 7.4), we performed additional CG simulations for Tau phase separation in the absence and presence of spermine (SPM) at different Tau:SPM ratio where we set the histidine charge to 0, characterizing fully deprotonated histidine at pH 7.4. From the simulations, the densities of the condensates are calculated and are compared in the following figure with the values obtained from our original simulations at pH 6.3. The new simulations reveal that while the absolute condensate density values shift to higher values under both spermine-free and spermine-containing conditions at pH 7.4, the qualitative trends observed, particularly regarding spermine’s ability to promote condensate formation, remained similar to our initial results at pH 6.3. This suggests that the slight acidification in the NMR experiments may modulate the absolute strength of electrostatic interactions or condensate density, but it does not alter the overall conclusions regarding spermine's effect on phase separation, especially for Tau. We have now added the **Fig. S19 in SI** to support this observation and the following text has been added in the revised manuscript:

"It is important to note that the CG simulation results discussed here correspond to the pH 6.3 conditions of the NMR experiments shown in Fig. 2D. We assigned a charge of +0.5 to histidine residues in the simulations, reflecting their partial protonation at pH 6.3 based on their pKa values. However, the in vitro and in vivo assays in our study were conducted at pH 7.4, where histidine

residues are expected to be fully deprotonated and thus carry a net charge of 0. This effect is more likely to impact Tau, which contains 12 histidine residues, but is less relevant for α S, which has only one. To assess whether the observations from our simulations remain valid under these conditions, we performed additional simulations of Tau phase separation, both in the absence and presence of spermine, using a histidine charge of 0 for pH 7.4. The condensate densities obtained from these simulations are compared in Fig. S19A with those from our original simulations at pH 6.3. The results show that, although the absolute condensate densities shift to higher values at pH 7.4, the qualitative trends, particularly regarding spermine's ability to promote phase separation, remain consistent across the Tau-spermine ratios examined. These findings suggest that while slight acidification in the NMR experiments may influence the magnitude of electrostatic interactions or condensate density, it does not affect the overall conclusions regarding spermine's role in promoting Tau phase separation." (Page 14, Line 396-413)

Figure S19. Tau phase separation at different pHs from coarse-grained simulations. (A) The densities of the Tau condensates at different pHs. (B) the histogram of linear fitting slope to the density-spermine concentration curve using a bootstrap test.

4.10. In vivo degradation kinetics

In the worm experiments, do the authors know how long spermine-treated condensates persist before clearance? Even approximate degradation kinetics could contextualize the biological relevance.

Answer: We thank the reviewer for this thoughtful suggestion. Based on our time-lapse imaging analysis (Fig. S20) shows that the number of inclusions in spermine-treated animals decreased over the observed period compared to control group (days 7–15). These observations support the spermine mediated α S condensates degradation via autophagy pathway. We have added this point to the revised Results section regarding condensate stability.

“However, with the spermine treatment, α S accumulation is significantly reduced in comparison to the control group at the same stage of adulthood (Fig. 4A-B & Fig. S20).” (Page 16, Line 461-463)

Figure S20: Degradation kinetics of α S condensates in *C.elegans* with and without spermine.

4.11. 1,6-Hexanediol control

The supplementary data **show sensitivity to 1,6-HD**, but it would help to mention this in the main text, as it supports the material state of the condensates.

Answer: We thank the reviewer for this helpful suggestion. Accordingly, we have revised the main text to explicitly mention these results.

“In the presence of 10% 1,6-hexanediol, Tau droplets formed without spermine were completely dissolved, while those formed with spermine exhibited partial resistance to dissolution and the number of droplets was reduced (Fig. 1F).” (Page 4, Line 136-139)

“1,6-Hexanediol was used to assess hydrophobic interactions, and the results demonstrated that 10% 1,6-hexanediol completely dissolved α S droplets, both in the presence and absence of spermine (Fig. 1L).” (Page 5, Line 168-170)

1. We thank the reviewer for highlighting this important point. In the revised Discussion, we now state that the new NMR data were generated only for Tau, and that our insights into α -synuclein rely in part on prior published studies (Fernández CO, et al. EMBO J. 2004 May 19;23(10):2039-46). Because we don't get the permission to adapt the study from EMBO journal, we have not made a schematic comparison.

To strengthen the comparative perspective, we have summarized the parallels and differences in spermine interactions with Tau and α -synuclein in Discussion section.

“Our insights into α S rely in part on prior NMR data²⁶, which indicated predominant specific interactions between the CTT of α S and spermine. In contrast, our new NMR data on Tau showed a more uniformly distributed residue-level interaction pattern.”

2. We appreciate the reviewer's suggestion to further clarify our interpretation. In the revised Discussion, we have stated that *“Our data are consistent with the hypothesis that liquid-like condensates are more accessible to autophagic clearance than fibrillar assemblies, although they do not provide conclusive proof.”*